# F1-Reasoner: Synthesizing Verifiable Reasoning Data From Formal Math Statements

## Abstract

Recent progress in reinforcement learning with verifiable rewards (RLVR) has substantially advanced the mathematical reasoning ability of large reasoning models (LRMs). However, existing datasets either rely heavily on manual annotation or are synthesized within artificial environments such as logic games. In this work, We propose a data synthesis framework that transforms formal mathematical statements into high-quality verifiable reasoning data. It first performs Statement Collection and Quality Control to obtain high-quality proven statements, then applies Problem Generation to convert them into verifiable math solving problems, and finally leverages RLVR with a verifier for Model Training. Using this framework, we synthesize 19k high-quality mathematical problems at levels 5–10 and train the F1-Reasoner series of models. Across six challenging benchmarks, F1-Reasoner consistently improves upon 3 different open-weight models across different sizes, outperforming models such as SynLogic and Absolute-Zero that are trained on verifiable data from other environments. Moreover, we mix our data with MATH to create F1-Reasoner-Mix, which further boosts performance; notably, F1-Reasoner-Mix-8B surpasses General-Reasoner-14B while using substantially less data. Further analysis shows that F1-Reasoner generalizes to informal theorem proving and exhibits richer thinking behaviors.

## 1 Introduction

Mathematical reasoning ability has achieved substantial improvements with the success of large reasoning models (LRMs) such as OpenAI-O1 (Jaech et al., 2024) and DeepSeek-R1 (Guo et al., 2025). These improvements are largely attributed to the System-2 thinking paradigm induced by reinforcement learning with verifiable rewards (RLVR) (Lambert et al., 2024; Luong et al., 2024). Subsequent studies such as SimpleRL (Zeng et al., 2025), DeepScaleR (Luo et al., 2025) and Logic-RL (Xie et al., 2025) have further shown that the effectiveness of RLVR is driven by high-quality verifiable reasoning data with powerful base models, which can even outperform the post-training techniques that includes process reward model (Lightman et al., 2024) and self-training (Zelikman et al., 2022).

However, constructing such mathematical reasoning data remains a major bottleneck, as it typically relies on manual annotation. For instance, SimpleRL utilizes math problems of level 3 to 5 from the MATH dataset (Hendrycks et al., 2021), while DeepScaleR and Polaris (An et al., 2025) leverage a large collection of previously published human math competition problems (MAA, a;b). As manually curated resources are depleted, the key challenge in advancing mathematical reasoning ability is synthesizing high-quality verifiable reasoning data (Ma et al., 2025; Zhang et al., 2025a).

Extensive studies has emerged for synthesizing verifiable reasoning data to strengthen mathematical reasoning capabilities. Early attempts primarily rely on rule-based augmentation of seed mathematical problems (Polozov et al., 2015; Koncel-Kedziorski et al., 2016; Kumar et al., 2022), which merely generate semantically similar variants. To generate diverse mathematical problems, subsequent works apply large language models (LLMs) to propose new questions (Yu et al., 2023; Tang et al.; Li et al., 2024), this approach is capable of synthesizing varied and novel problems. However, ensuring correctness in synthetic complex problems remains a major challenge and is particularly critical in RLVR. Motivated by these challenges, recent efforts focus on construct reasoning data within verification environments such as diverse logic games (Liu et al., 2025; Helff et al., 2025) or

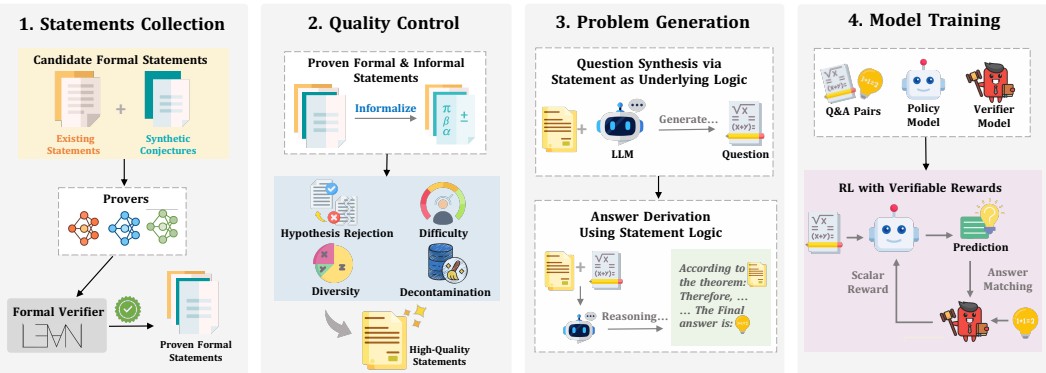

Figure 1: Overview of the F1-Reasoner development pipeline, which consists of Statements Collection, Quality Control, Problem Generation and Model Training.

code reasoning tasks (Zhao et al., 2025). They can synthesize correct and complex verifiable reasoning data, but the gap still exists between such environments and mathematical reasoning tasks.

In this paper, we explore the potentiality of synthesizing verifiable reasoning data from formal theorem proving systems. This proposal is inspired by following insights: (1) Formal theorem proving systems utilize verifiers like Lean (De Moura et al., 2015) or Isabelle (Nipkow et al., 2002) to verify mathematical statements, thus ensuring the quality of the synthetic data. (2) Neural theorem proving has made significant progress recently, with IMO-level mathematical statements now being automatically proved (AlphaProof and AlphaGeometry teams, 2024; Chen et al., 2025). However, synthesizing verifiable reasoning data from formal theorem proving systems faces two main challenges. On the one hand, how to design an effective synthesis pipeline to generate high-quality reasoning data; on the other hand, how to convert mathematical statements into verifiable question-answer pairs (Zhang et al., 2025b; Sheng et al., 2025).

To address these challenges, we introduce a data synthesis framework for generating high-quality verifiable reasoning data from formal math statements. Using the proposed framework, we synthesize 19k high-quality math reasoning data at levels 5–10 [1] and obtain the **F1-Reasoner** model series trained on this dataset. As shown in Figure 1, it consists of four components. **Statement Collection** gathers existing theorems and synthetic conjectures, retaining only those verified by a formal proving system. **Quality Control** applies multiple procedures to filter the statements, including hypothesis rejection, difficulty assessment, and diversity sampling. **Problem Generation** leverages the underlying logic of each statement to synthesize related mathematical problems, with answers derived directly from the statements to ensure verifiability and correctness. **Model Training** involves training models in RLVR with a verifier checking predictions in comparison with the answers.

To validate the effectiveness of our approach, we conduct experiments on 6 challenging mathematical reasoning benchmarks. F1-Reasoner demonstrates consistent improvements across multiple base models, showing clear advantages over approaches trained on verifiable data synthesized from other environments such as SynLogic and Absolute-Zero. Furthermore, by combining our synthesized data with problems from MATH, we train a series of **F1-Reasoner-Mix** models. Its performance is further strengthened, with F1-Reasoner-Mix-8B even surpassing General-Reasoner-14B despite using significantly less data. We further observe that F1-Reasoner's reasoning ability can generalize to informal theorem proving. Finally, a detailed analysis of F1-Reasoner's reasoning chains reveals that it exhibits richer thinking behaviors.

## 2 F1-REASONER

The construction of F1-reasoner consists of four main components. First, we collect formal mathematical statements that have been successfully proven by the formal theorem proving system. Sec-

---

[1]We are referencing the *AoPS Wiki Competition Ratings* website for this assessment.

ond, we apply quality control procedures to select high-quality statements. Third, we synthesize verifiable mathematical question–answer pairs by using statements as underlying logic. Finally, we employ RLVR to train the model on synthetic dataset. We will provide a detailed description of Statements Collection in Section 2.1, Quality Control in Section 2.2, Problem Generation in Section 2.3, and Model Training in Section 2.4.

## 2.1 STATEMENTS COLLECTION

In neural (formal) theorem proving systems, mathematical statements are first expressed in formal mathematical languages, such as Lean (De Moura et al., 2015) or Isabelle (Nipkow et al., 2002). Subsequently, the neural prover attempts to generate multiple verifiable candidate formal proofs for each statement. A statement is regarded as proven if at least one of its proofs successfully passes the proof-checking compiler.

In the statements collection phase, we aim to obtain a large set of correct mathematical statements for subsequent problem generation. As shown in Figure 1, we first collect a large pool of candidate formal math statements. Then we retain only those statements that are successfully validated by the formal theorem proving system. This procedure provides two key advantages: (1) Robustness. The correctness of math statements is guaranteed by formal verification through the compiler. (2) Powerfulness. As open-source automated provers become stronger, they can produce correct proofs for challenging problems (Dong & Ma, 2025; Ren et al., 2025; Wu et al., 2024; Lin et al., 2025). Specifically, among the candidate formal statements, we include those from Lean-Workbook (Ying et al., 2024), which were translated from natural language problems by auto-formalisation models, as well as synthetic conjectures generated by the STP conjecturer (Dong & Ma, 2025) based on existing formal statements. Finally, we retain statements that are successfully proven by either InternLM2.5-StepProver (Wu et al., 2024) or STP-Prover (Dong & Ma, 2025), resulting in a total of 740,447 proven formal statements.

## 2.2 QUALITY CONTROL

Data quality is a key factor in training large language models, as high-quality data can result in a less-is-more effect (Ye et al., 2025; Li et al., 2025). Therefore, we conduct following quality control on our proven formal statements to select the most high-quality subset.

**Hypothesis Rejection.** In collected proven formal statements, although provable, some of them are based on inconsistent hypotheses and result in vacuous conclusions. For example, the statement in Figure 2 includes inconsistent hypotheses between $f(x) = \left(\frac{x}{a}\right)^2 + \frac{x}{a} + 1$ and $f(a) = 0$, which makes subsequent conclusions meaningless. To filter out such cases in our dataset, we utilize the hypothesis rejection strategy (Xin et al., 2024). It uses a prover model to attempt proving the given statement with 'False' as the conclusion . Any statement that can be successfully proven is identified as invalid.

---

theorem main_2016 (a : ℝ) (h₁ : a > 0)
  (h₂ : ∃ f : ℝ → ℝ, ContinuousOn f (Set.Icc 0 a) ∧
    ∀ x ∈ Set.Icc 0 a, f x = (x / a) ^ 2 + x / a + 1 ∧ f a = 0) :
  ∃ x ∈ Set.Icc 0 a, x * x > √2 - √5

                                            *f a = (a / a) ^ 2 + a / a + 1*
                                            *= 3 ≠ 0* ***Contradiction !***

theorem main_2016 (a : ℝ) (h₁ : a > 0)
  (h₂ : ∃ f : ℝ → ℝ, ContinuousOn f (Set.Icc 0 a) ∧
    ∀ x ∈ Set.Icc 0 a, f x = (x / a) ^ 2 + x / a + 1 ∧ f a = 0) : False := by

---

Figure 2: A statement with inconsistent hypotheses and its corresponding 'False' conclusion version for hypothesis rejection strategy.

To further minimize the presence of incorrect data, we also employ an LLM-as-judge strategy to check whether the statements contain contradictions (the detailed prompt in Appendix D.1). A statement is discarded if it is detected by the hypothesis rejection strategy or identified by the LLMs.

**Difficulty.** For advancing mathematical reasoning of LLMs, high difficulty problems are essential as it can improve both the performance on challenging benchmarks and generalization (Shao et al., 2024; Muennighoff et al., 2025). Therefore, we aim to examine whether our proven formal statements contain sufficiently difficult problems and extract them for training.

To achieve a more objective assessment of problem difficulty, we follow Omni-Math (Gao et al., 2024) to leverage LLM judgment referencing the *AoPS Wiki Competition Ratings* website, which assign difficulty scores from 0 to 10 across a wide range of competition problems. Surprisingly, we find that over 100k proven statements are rated at level 5 and above. It indicates that current formal theorem proving systems can already successfully prove difficult problems, highlighting its potential as a tool for synthesizing verifiable and challenging data. We provide the detailed difficulty statistics and judgment prompt in Appendix D.2. For subsequent stages, we retain only statements with difficulty $\geq 5$ for further filtering.

**Diversity.** Among the filtered statements, we observe that numerous statements are concentrated at same difficulty levels and exhibit high semantic similarity. To improve both training efficiency and performance, we perform diversity sampling. Specifically, for each difficulty level, we adopt the Deita (Liu et al., 2023b) data selection pipeline: (1) **Complexity Ranking**. Within the same difficulty level, we apply the provided scorer[2] to assign complexity scores and rank them in descending order. (2) **Diversity Sampling**. Starting from the most complex statements, we iteratively add candidate statements to the data pool. A candidate is included only if its similarity with all statements in the pool is below a defined threshold. Finally, this process yields over 19k high-quality statements.

**Decontamination.** At the statement collection stage, some statements are translated from existing natural language problems by the community. Since this process may introduce potential data contamination, we conduct careful contamination detection and filtering.

Specifically, our detection pipeline consists of three steps: (1) **Test-Set Collection**. We collect a test pool from widely used math benchmarks such as AIME, AMC, MATH, Minerva, and Olympiad-Bench. (2) **Contamination Candidates Retrieval**. For each statement, we retrieve its top-5 most similar problems from the test pool. (3) **Contamination Identification.** Each statement is compared with its retrieved candidates using LLMs to determine whether they are the same problem. If so, the statement is discarded. In the end, we obtain a final set of 19,112 high-quality statements.

## 2.3 PROBLEM GENERATION

Converting the collected high-quality statements into question–answer pairs has two challenges. First, a statement usually expresses an objective mathematical fact rather than a problem formulation, making it difficult to extract a question–answer pair; if forced into the form of a proof problem, its corresponding proof is often difficult to verify (Zhang et al., 2025b; Sheng et al., 2025). Second, when generating questions about the given statement, obtaining correct and reliable answers is challenging. To address these challenges, we propose a novel Problem Generation method including Question Synthesis and Answer Derivation. We will provide a detailed description as follows.

**Question Synthesis.** We synthesize questions by taking each statement as underlying logic. Rather than directly asking the model to prove the statement, we treat it as background knowledge within the mathematical world and generate a new question based on it. Figure 3 shows that while the statement describes an inequality, the synthesized question requires finding numbers that satisfy the reverse inequality. Solving such questions requires the model to first analyze the question, identify the underlying logic and reason toward the final answer. Figure 3 illustrates this process, where GPT5-mini performs extensive reasoning (1,899 tokens) before arriving at the correct answer. To better demonstrate our generation principle as well as the diversity and generalization of the generated questions, we present additional cases in the Appendix B.

**Answer Derivation.** For difficult question, obtaining the answer solely through the model's reasoning ability is challenging. To address this, we propose directly providing the model with the above statement and instructing it to use it to solve the synthetic question. This transformation greatly simplifies the problem, as the model only needs to apply the theorem to obtain the correct answer, thereby ensuring reliability and correctness. In the overall problem generation process, we gener-

---

[2]An LLM-based scoring model publicly released by Deita.

**Collected High-Quality Statement:**

**Formal Statement:**
theorem lean_workbook_plus_75540 (a : ℝ) (ha : 0 < a) : 16 * (a + 2)^4 * (2 * a + 1) ≥ 243 * (a + 1)^4  := by sorry

**NL Statement:**
For any positive real number a , the expression 16 times (a + 2)^4 times (2a + 1) is greater than or equal to 243 times (a + 1)^4.

**The Details of Problem Generation:**

**Question Synthesis:** Let A be a positive real number. Find all values of  A for which the inequality  16 times (A + 2)^4 times (2A + 1) < 243 times (A + 1)^4  holds. ↘ **Generating an inequality opposite to the original statement.**

**Answer Derivation:** Consider the given inequality: 16 times (A + 2)^4 times (2A + 1) < 243 times (A + 1)^4. We are to find all positive real numbers A such that the left side is strictly less than the right side. However, by analyzing the given theorem, we note that the left side is always greater than or equal to the right side for any positive real A. This means that there are no positive real values of A for which the inequality is satisfied. ↘

**Directly deriving the correct answer using the original statement.**

**Final Answer:** No such A exists.

**The Reasoning Process of LLMs Trying to Solve the Synthetic Question:**

⊛ **GPT5-mini**:
Let's solve the inequality carefully step by step. Step 1: Rewrite the inequality in a nicer form
… Step 2: Simplify the constants…Step 3: Make a substitution …Step 4: Solve the inequality for x …Step 5: Factor if possible….Step 6: Solve numerically …Test some values… **So seems inequality never holds for A>0 !**
Conclusion: The inequality: 16(A+2)4(2A+1)<243(A+1)416 (A+2)^4 (2A+1) < 243 (A+1)^4 has no positive solutions.
**Final Answer: No positive A satisfies the strict inequality.**                    **(A total of 1,899 Tokens)**

Figure 3: A case of Problem Generation and the reasoning process of LLMs on it.

ate one question–answer pair for each statement. The whole problem generation instruction is in Appendix D.3.

## 2.4 MODEL TRAINING

To train the policy model, we adopt the reinforcement learning with verifiable rewards. Given a question–answer pair $(x, y^*)$, the policy $\pi_\theta$ predicts an answer $y$, which is compared with ground truth $y^*$ to produce a scalar reward $R(x, y, y^*)$. The training objective is to maximize the expected reward:

$$J(\theta) = \mathbb{E}_{x,y \sim \pi_\theta}[R(x, y, y^*)],$$

and the gradient of the objective can be estimated as:

$$\nabla_\theta J(\theta) = \mathbb{E}_{x,y \sim \pi_\theta}\left[\nabla_\theta \log \pi_\theta(y|x)(R(x, y, y^*) - b(x))\right],$$

where $b(x)$ is a baseline used to reduce variance. In practice, we employ GRPO (Shao et al., 2024) as the optimization algorithm.

A key distinction from traditional rule-based answer checking is that our dataset contains answers in diverse formats such as strings and structured expressions. To handle this, we apply a verifier model $V_\phi$ that outputs a binary correctness signal $V_\phi(x, y^*, y) \in \{0, 1\}$, indicating whether $y$ is equivalent to $y^*$. The reward function is then simply defined as $R(x, y, y^*) = V_\phi(x, y^*, y)$. Concretely, we employ the open-source General-Verifier (Ma et al., 2025) as our verifier, this training recipe enables efficient policy learning on our synthetic dataset.

## 3 EXPERIMENTS

### 3.1 EXPERIMENT SETUP

**Training Details.** We follow the Zero RL setting, directly conducting reinforcement learning from base large language models without an intermediate supervised fine-tuning stage. Specifically, we initialize our models with Qwen2.5-7B (Qwen, 2025) and Qwen3-4B/14B (Yang et al., 2025). Our

Table 1: Comprehensive results on mathematical reasoning benchmarks. The peak performance achieved during each model's training process is highlighted in **bold**.

| Model Name | AVG. | AMC | Minerva | MATH | Olympiad | AIME25 | AIME24 |
|---|---|---|---|---|---|---|---|
| *Qwen2.5-7B-Base* | | | | | | | |
| Base Model | 26.67 | 30.00 | 36.00 | 60.20 | 28.60 | 1.40 | 3.80 |
| Qwen2.5-7B-Instruct | 38.85 | 52.50 | 45.20 | 75.00 | 39.40 | 8.50 | 12.50 |
| SynLogic | 35.94 | 47.66 | 42.28 | 71.80 | 38.66 | 5.21 | 10.00 |
| Absolute-Zero | 35.82 | 52.50 | 38.20 | 73.20 | 38.50 | 4.16 | 8.33 |
| F1-Reasoner | 38.15 | 48.00 | 52.90 | 74.40 | 38.10 | 3.80 | 11.70 |
| SimpleRL-Zoo | 41.43 | **60.00** | 49.60 | 74.00 | **41.90** | 7.50 | **15.60** |
| General-Reasoner | 41.18 | 55.00 | **54.00** | 76.00 | 37.90 | **10.40** | 13.80 |
| F1-Reasoner-Mix | **41.62** | 54.50 | 52.90 | **78.00** | 40.70 | 8.80 | 14.80 |
| *Qwen3-4B-Base* | | | | | | | |
| Base Model | 19.70 | 27.50 | 18.40 | 38.80 | 16.40 | 6.15 | 10.94 |
| Qwen3-4B (non-think) | 47.92 | 62.50 | 57.00 | 80.40 | 49.00 | 16.10 | **22.50** |
| General-Reasoner | 46.90 | 60.00 | **57.70** | 80.60 | 47.70 | 15.40 | 20.0 |
| F1-Reasoner | 40.80 | 49.10 | 56.20 | 78.00 | 42.70 | 7.10 | 14.10 |
| F1-Reasoner-Mix | **49.53** | **66.60** | 56.20 | **84.20** | **50.00** | **18.00** | 21.80 |
| *Qwen3-8B-Base* | | | | | | | |
| Base Model | 42.54 | 51.95 | 50.00 | 78.00 | 44.74 | 16.67 | 13.85 |
| Qwen3-14B (non-think) | 50.23 | 57.50 | 55.90 | 82.00 | 52.40 | 25.10 | 28.50 |
| General-Reasoner-14B | 52.88 | 70.00 | 68.00 | 83.80 | 51.90 | 19.20 | 24.40 |
| SynLogic | 44.72 | 55.00 | 60.30 | 80.00 | 46.40 | 13.90 | 12.70 |
| F1-Reasoner | 46.20 | 60.80 | 59.60 | 80.60 | 40.70 | 12.20 | 20.00 |
| SimpleRL-Zoo | 53.78 | **75.90** | 52.60 | **88.20** | 51.90 | **23.30** | **30.80** |
| F1-Reasoner-Mix | **54.42** | 72.30 | **66.90** | 87.60 | **55.60** | 20.40 | 25.10 |

reward function employs a binary scoring mechanism that combines a format reward and an answer reward. For training data, we consider two settings: F1-Reasoner uses only our synthesized dataset containing 19,112 reasoning QA pairs. F1-Reasoner-Mix combines this with 8,920 Level 3–5 problems from MATH dataset. For selecting training checkpoints, we use the average accuracy over 16 runs of AIME 2024 as our validation set. Our implementation is based on the verl repository [3].

**Evaluation.** We evaluate our models on six challenging problem-solving benchmarks: AMC 23 (MAA, b), Minerva (Lewkowycz et al., 2022), MATH-500 (Hendrycks et al., 2021), Olympiad-Bench (He et al., 2024), AIME-2024 and AIME-2025 (MAA, a). For the AMC and AIME benchmarks, we report the mean@32 metric with sampling temperature set to 1, in order to ensure more robust evaluation. For all other benchmarks, we report accuracy under greedy decoding. To assess correctness, we use the simple-evals [4] evaluation framework, and combine rule-based matching and GPT-4o (OpenAI, 2024) to ensure robust verification of answer correctness.

**Baselines.** Our main baselines are grouped into two categories: 1) Synthetic Data. These methods synthesize verifiable reasoning data in artificial environments. SynLogic (Liu et al., 2025): It constructs 49k verifiable logic data from 35 different logic games. Absolute-Zero (Zhao et al., 2025): It generates verifiable reasoning tasks in code environments and employs a curriculum learning strategy that enables the model to gradually self-evolve towards more complex reasoning. 2) Human Data. These methods rely on human-curated math data to train reasoning models. SimpleRL (Zeng et al., 2025): It directly train the models on MATH Level 3–5 problems. General Reasoner (Ma et al., 2025): It extracts 230k diverse verifiable reasoning samples from large-scale pre-training corpora, aiming to cover a broad range of diverse reasoning scenarios.

## 3.2 MAIN RESULTS

**F1-Reasoner outperforming data synthesized in other environments.** As shown in Table 1, training on our synthetic data consistently enhances the performance of Qwen2.5-7B-Base, Qwen3-

---

[3] https://github.com/volcengine/verl
[4] https://github.com/openai/simple-evals

4B-Base, and Qwen3-8B-Base. On Qwen2.5-7B, F1-Reasoner significantly exceeds Synlogic (by 2.21%) and Absolute-zero (by 2.33%) on average score. Notably, on Minerva-Math, F1-Reasoner-7B achieves 52.9, surpassing Synlogic by 10.62%, Absolute-zero by 14.7%, and even exceeding SimpleRL-Zoo by 3.3%. It highlights the importance of synthesizing verifiable reasoning data in formal math environments for strengthening mathematical reasoning ability. F1-Reasoner-7B which is only trained on synthetic data, achieves competitive performance with Qwen2.5-7B-Instruct, which is post-trained with a large mixture of human-annotated and synthetic data. Furthermore, we reproduce SynLogic on Qwen3-8B for a fair comparison. We observe that on Qwen3-8B, our F1-Reasoner outperforms SynLogic by 1.48 points on average. The consistent superiority demonstrates that our generated data remains effective even on stronger backbones. These results show the effectiveness of our synthetic data and data generation pipeline.

**Human data combining our data further improves performance.** To verify whether human data can further enhance math reaosning capabilities when merged with our synthetic data, we selected SimpleRL-Zoo training data (MATH level 3–5) as human-annotated data and trained F1-Reasoner-Mix. We observe that F1-Reasoner-Mix consistently outperforms SimpleRL-Zoo across both Qwen2.5-7B and Qwen3-8B, demonstrating the efficacy of our synthetic data. Specifically, our method achieves improvements of approximately X and Y points on these models, respectively. Notably, on Qwen3-8B, F1-Reasoner-Mix maintains robust performance on general benchmarks while delivering a substantial breakthrough on Minerva-Math (+14.3 points). Given that Minerva-Math emphasizes scientific and quantitative reasoning (e.g., astronomy, physics), we attribute this gain to the diverse theorem-based knowledge within our F1 dataset, which effectively enhances the model's generalization capabilities in scientific domains. Remarkably, comparing to the General-Reasoner which trained on 230k collected human reasoning data, F1-Reasoner-Mix exceeds the Qwen2.5-7B General-Reasoner by 0.44%, the Qwen3-4B General-Reasoner by 2.63%, and even surpasses the Qwen3-14B General-Reasoner by 1.54% on Qwen3-8B. Furthermore, F1-Reasoner-Mix also outperforms the corresponding instruct models, including Qwen2.5-7B-Instruct, Qwen3-4B-Instruct (non-think) and even Qwen3-14B-Instruct (non-think). This finding demonstrates that when human-annotated data is mixed with our synthetic data, it can activate the complex reasoning abilities of models. Moreover, the effectiveness of our challenging synthetic mathematical data are more evident with stronger base models.

## 3.3 FINE-GRAINED ANALYSIS

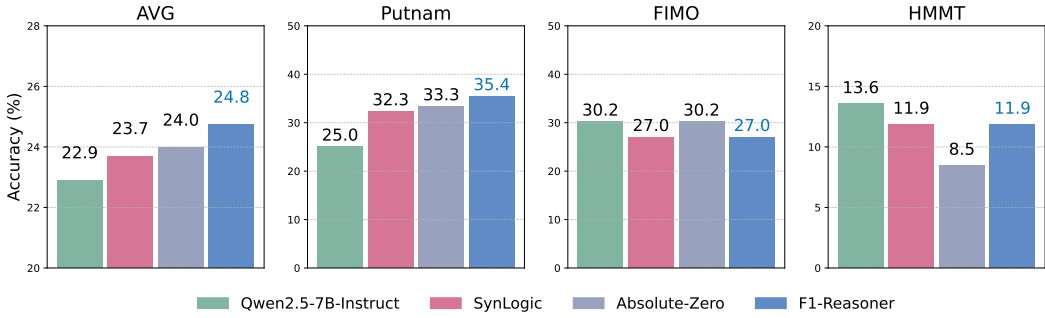

Figure 4: The 7B models performance on Putnam, FIMO and HMMT.

**Model Performance on Informal Proving.** Since our data are generated from formal theorem statements, we try to investigate whether F1-Reasoner can generalize to informal theorem proving. To evaluate the model's performance in informal proving, we followed the DeepTheorem (Zhang et al., 2025b) evaluation protocol. Specifically, DeepTheorem collects seed statements from Putnam-Bench (Tsoukalas et al., 2024), FIMO (Liu et al., 2023a) and HMMT (Harvard–MIT Mathematics Tournament, 2024–2025), then manually constructs their entailing variants and contradictory variants. The model is asked to directly judge whether each given statement is correct, and a problem is considered correctly proved only if the model correctly classifies both the original statement and all its variant statements. As shown in Figure 4, F1-Reasoner achieves an average score of 24.8, surpassing Qwen2.5-Instruct-7B, Synlogic, and Absolute-Zero. On PutnamBench, F1-Reasoner attains

the best performance among all models, exceeding Absolute-zero by 1.9%, Synlogic by 3.1% and Qwen2.5-Instruct-7B by 10.4%. And F1-Reasoner demonstrates competitive performance on FIMO and HMMT as well. These results indicate that models trained on data synthesized in our method not only improve mathematical reasoning in problem-solving tasks but also generalize effectively to informal theorem proving.

**Training on Data without Formal System Verification.** Recently, some works focus on generating math reasoning data only using LLMs. These approaches can generate correct but intermediate problems, their correctness is hard to guarantee as difficulty increases. To investigate the importance of correctness in difficult problems for RL, we conducted a comparative experiment using data not verified by a formal theorem proving system. Specifically, we collect the final 19k statements from our pipeline and then prompt GPT-4o to generate new statements that mimic but are not identical to them. This yields additional similar statements without formal verification. After that, we directly use them in subsequent problem generation and produce 19k QA pairs. We train on Qwen3-8B-Base by mixing this data with the MATH Level3-5 data, in order to examine the potential harm introduced by such unreliable data.

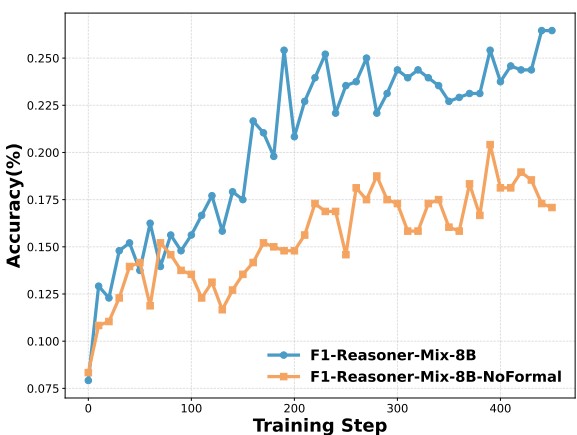

Figure 5: Training dynamics of data without without formal verification. The accuracy represents the mean performance of 16 runs on the AIME 2024.

As shown in Figure 5, incorporating these unreliable data leads to two major issues: (1) **Early Saturation.** The learning curve of F1-Reasoner-Mix-8B-NoFormal is initially similar with F1-Reasoner-Mix-8B before Step-80. However, its performance quickly saturates and only increases very slowly after that, resulting in poor training efficiency. In contrast, F1-Reasoner-Mix-8B continues to improve rapidly until Step-200 and maintains steady gains thereafter. (2) **Suboptimal Performance.** F1-Reasoner-Mix-8B-NoFormal reaches the maximum score of only 20.42% at Step-390, whereas F1-Reasoner-Mix-8B achieves the score of 26.67%. It demonstrates that data correctness is a critical factor in RL training and highlights the importance of formal checking environment in synthetic math problems. We guess that model's initial rapid performance gains reflect adaptation to the output format, whereas further gains depend on the quality and correctness of the training data.

**Reasoning Behavior of F1-Reasoner.** To better understand the reasoning behaviors of F1-Reasoner, we conducted a detailed analysis of their output length, diversity and frequency of reasoning skills. Specifically, we collect responses from 16 runs AIME24 and adopt the cognitive behavior framework proposed by Gandhi et al. (2025), which identifies reasoning-related behaviors such as *Backtracking* and *Verification* by LLMs. We then statistic both the diversity and overall frequency of reasoning skills. Figure 6 and Figure 7 present the results, our analysis reveals two main findings: (1) **F1-Reasoner employs numerous reasoning skills within Short Reasoning Traces.** As shown in Figure 6, Average reasoning length of F1-Reasoner is only 784 tokens, which is the shortest among all models. Despite its short length, F1-Reasoner produces 1200 reasoning behaviors in total, comparable to SimpleRL-Zoo's 1257 but using far fewer tokens (1968). Furthermore, combining SimpleRL-Zoo's data with ours to form F1-Reasoner-Mix greatly reduces reasoning length, improving efficiency while retaining reasoning behaviors (1202). (2) **Richer Reasoning Skills.** As illustrated in Figure 7, F1-Reasoner demonstrates the most diverse reasoning skills. We attribute this effect to the nature of our generated data. Many of our problems require multiple reasoning skills, such as enumeration and backtracking, to uncover implicit theorem knowledge and solve them. Consequently, training on such problems allows the model to explore a richer set of reasoning skills at inference time. In addition, by learning which skills succeed for different problem types, the model can select more effective strategies, leading to the efficient reasoning. A similar phenomenon

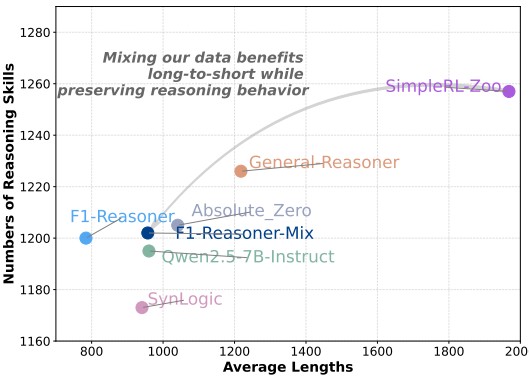

Figure 6: The numbers of reasoning behavior and the average lengths of responses on 7B models.

| Model | Reasoning Skills |
|---|---|
| Qwen2.5-7B-Instruct | |
| SynLogic | |
| Absolute-Zero | |
| F1-Reasoner | |
| General-Reasoner | |
| SimpleRL-Zoo | |
| F1-Reasoner-Mix | |

Figure 7: Reasoning skills diversity across 7B models. ■: Backtracking, ■: Enumeration, ■: Subgoal Setting, ■: Verification, ■: Abstraction, ■: Generalization, ■: Pattern Recognition, ■: Symmetry, ■: Analogy Other Cases, ■: Application Theorems

is observed in Gandhi et al. (2025), where introducing more cognition reasoning data in continual pretraining led to more diverse reasoning behaviors.

## 4 RELATED WORK

**Mathematical Reasoning of LLMs.** Community has made significant efforts to enhance the mathematical reasoning ability of large language models (LLMs). In the pre-training stage, previous works focus on collecting or synthesizing large-scale math-related corpora (Wang et al., 2023; Paster et al., 2024; Shao et al., 2024; Akter et al., 2024). Moreover, Lin et al. (2024) proposed to select key tokens in math data to improve pre-training efficiency. In the post-training stage, numerous studies generated massive problem-solving data to conduct supervised fine-tuning (Yue et al., 2023; Luo et al., 2023; Tang et al., 2024). In addition, reinforcement learning methods such as GRPO (Shao et al., 2024) and PRM (Lightman et al., 2024) have been shown to further boost reasoning abilities. Recently, large reasoning models (Jaech et al., 2024; Guo et al., 2025) have shown that simple RLVR (Lambert et al., 2024; Zeng et al., 2025; Luong et al., 2024) already achieve remarkable performance with high-quality verifiable reasoning data. Therefore, synthesizing such high-quality QA pairs has become a critical issue.

**Verifiable Reasoning Data Synthesis.** Early efforts on data synthesis mainly relied on rule-based (Polozov et al., 2015; Kumar et al., 2022) or rewriting-based (Koncel-Kedziorski et al., 2016; Moon-Rembert & Gilbert, 2019) approaches to augment math word problems. However, such approaches are limited to semantic variations. With the advent of LLMs, numerous studies turn to LLM-based generation for generating math QA pairs (Yu et al., 2023; Yue et al., 2023; Li et al., 2024). While this paradigm significantly improves novelty and diversity, the correctness of generated anwers cannot be guaranteed. To address this challenge, recent works explore artificial environments such as logic games (Liu et al., 2025; Helff et al., 2025) or code reasoning tasks (Zhao et al., 2025), where the correctness of answer can be automatically verified. Nevertheless, these environments still lack the diversity of mathematical knowledge and logic structures of math problems.

**Neural Theorem Proving.** Previous neural theorem proving system was predominantly driven by tactic search methods (Blaauwbroek et al., 2024; Jiang et al., 2022; Lample et al., 2022; Wu et al., 2024). These approaches generate tactics step by step, guided by search algorithms such as heuristic search (Gauthier et al., 2018) and Monte-Carlo tree search (Xin et al., 2025). Such a tactic-search paradigm often scales poorly with increasing neural model size. Subsequently, with the emergence of long-chain-of-thought reasoning, whole proof generation has become increasingly competitive (Jiang et al., 2023; First et al., 2023). For example, Kimina-Prover (Wang et al., 2025) leverages long-CoT and reinforcement learning to achieve performance surpassing tactic-search systems. Recently, Seed-Prover (Chen et al., 2025) employed a lemma-style whole-proof-generation paradigm and successfully solved 5 of the 6 IMO 2025 problems. This demonstrates that modern formal provers are now powerful enough to address highly challenging competition-level problems.

## 5 CONCLUSION AND LIMITATION

**Conclusion.** In this work, we introduced **F1-Reasoner**, a framework for synthesizing high-quality verifiable reasoning data from formal mathematical statements. Our pipeline integrates statement collection, rigorous quality control, and problem generation to construct a dataset of 19k challenging problems. Building on this dataset, we trained the F1-Reasoner models with RLVR, achieving consistent gains across six math solving benchmarks. Both F1-Reasoner and its Mix version outperform baselines that rely on either synthetic data from artificial environments or human data. F1-Reasoner also generalizes to informal theorem proving and exhibits richer reasoning behaviors. Our work highlights the value of formal systems as a source of scalable and verifiable data.

**Limitation.** While our work takes a first step in synthesizing verifiable data from formal proving system, it has two primary limitations. First, while our current framework is a static pipeline, our vision is to build an self-improving loop where the model iteratively evaluates itself and generates targeted data like Absolute-Zero (Zhao et al., 2025). Second, due to computational constraints, our experiments are conducted on models up to 8B parameters, whereas other methods are validated on larger 14B (Ma et al., 2025) or 32B models (Liu et al., 2025). The full potential of our data on larger models remains unexplored. These limitations inform our future research directions.

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

## A  LARGE LANGUAGE MODEL USAGE STATEMENT

We acknowledge the use of Large Language Models in the preparation of this manuscript. Beyond their core role in our methodology for data generation and evaluation as detailed in the main paper, we utilized LLMs for two primary supporting tasks. First, we employed an LLM for polishing phrasing, improving clarity and correcting grammatical errors. Second, LLMs were used as a coding assistant for the implementation of certain parts of our project. The core intellectual contributions of this work, including the research idea, the overall structure and logical flow of the paper, and the literature review, were exclusively conducted by the human authors.

# B  CASES OF SYNTHETIC DATA

Here, we present several representative examples from our synthetic dataset, including their formal statements, informal statements, the generated questions and answers, as well as the reasoning processes that lead to the correct answers. In addition, we also provide their corresponding difficulty levels and the relevant mathematical domain knowledge involved.

---

**The Information of ID: 628531**

**Formal Statement**:
```
theorem lean_workbook_16724 (a b c :  ℝ) :
13 * Real.sqrt (a^2 + b^2 + c^2) ≥ 3 * a + 4 * b + 12 * c ∨
(Real.sqrt (a^2 + b^2 + c^2) - 12 * a) * (Real.sqrt (a^2 + b^2 + c^2) - 12 * b) + Real.sqrt
(12 * a * b + 6 * b * a) ≥ 0 := by
```

**Informal Statement**:
For any real numbers $a$, $b$, and $c$, either 13 times the square root of $a^2 + b^2 + c^2$ is greater than or equal to $3a + 4b + 12c$, or the expression $(\sqrt{a^2 + b^2 + c^2} - 12a)(\sqrt{a^2 + b^2 + c^2} - 12b) + \sqrt{12ab + 6ba}$ is greater than or equal to zero.

**Generated Question**:
Let $a$, $b$, and $c$ be real numbers that satisfy

$$13\sqrt{a^2 + b^2 + c^2} < 3a + 4b + 12c.$$

Determine whether the following statement is always true, always false, or sometimes true:

$$(\sqrt{a^2 + b^2 + c^2} - 12a)(\sqrt{a^2 + b^2 + c^2} - 12b) + \sqrt{12ab + 6ba} \geq 0.$$

**Answer Derivation Process**:
We are told $13\sqrt{a^2 + b^2 + c^2} < 3a + 4b + 12c$.
By the given theorem (though not stated in the problem), for any real $a, b, c$, at least one of the two conditions holds: 1. $13\sqrt{a^2 + b^2 + c^2} \geq 3a + 4b + 12c$, or 2. $(\sqrt{a^2 + b^2 + c^2} - 12a)(\sqrt{a^2 + b^2 + c^2} - 12b) + \sqrt{12ab + 6ba} \geq 0$.
Since in our problem statement it is given that condition (1) does **not** hold (the left side is strictly less), the theorem ensures that condition (2) **must** hold.
Therefore: For all real numbers $a, b, c$ satisfying the given inequality, the second expression is guaranteed to be greater than or equal to zero.

**Generated Answer**: Always true.

**Difficulty**:
5.5

**Domain**:
['Algebra', 'Intermediate Algebra', 'Inequalities'], ['Algebra', 'Linear Algebra', 'Vectors and Norms'], ['Applied Mathematics', 'Other']

---

864
865
866
867
868
869
870
871
872
873
874
875
876
877
878
879
880
881
882
883
884
885
886
887
888
889
890
891
892
893
894
895
896
897
898
899
900
901
902
903
904
905
906
907
908
909
910
911
912
913
914
915
916
917

**The Information of ID: 382378**

**Formal Statement**:
```
theorem lean_workbook_8896 (x y z :  ℝ)
(h4 :  x^4 + y^4 + z^4 = 3) (h3 :  x^3 + y^3 + z^3 = 3) (h2 :  x^2 + y^2 + z^2 = 3) :
x + y + z = 3 := by
```

**Informal Statement**:
Suppose $x$, $y$, and $z$ are real numbers such that the sum of their fourth powers is 3, the sum of their cubes is 3, and the sum of their squares is 3. Then the sum of $x$, $y$, and $z$ must be 3.

**Generated Question**:
Three real numbers $a, b, c$ are such that:

$$a^4 + b^4 + c^4 = 3, \quad a^3 + b^3 + c^3 = 3, \quad a^2 + b^2 + c^2 = 3.$$

Using this information, determine the exact value of $a + b + c$.

**Answer Derivation Process**:
We are given the three conditions:

$$a^4 + b^4 + c^4 = 3, \quad a^3 + b^3 + c^3 = 3, \quad a^2 + b^2 + c^2 = 3.$$

The hidden theorem ensures that when these three equalities hold simultaneously, the value of the sum of the three numbers, $a + b + c$, is uniquely determined.
By invoking that theorem, we conclude:
$$a + b + c = 3.$$

**Generated Answer**: 3

**Difficulty**:
5.5

**Domain**:
['Algebra', 'Intermediate Algebra', 'Symmetric Polynomials'], ['Algebra', 'Intermediate Algebra', 'Polynomial Identities']

**The Information of ID: 117842**

**Formal Statement**: `theorem exercise_10_4_51 (M : Type*) [CommRing M] [NoZeroDivisors M] (A : Ideal M) (hA : A * A = ⊥) :  A = ⊥ := by`

**Informal Statement**:
Let $M$ be a commutative ring with no zero divisors, and let $A$ be an ideal of $M$. If the product of $A$ with itself is the zero ideal, then $A$ must also be the zero ideal.

**Generated Question**:
Let $R$ be a commutative ring with no zero divisors. Consider an ideal $I \subseteq R$ such that every product of two elements from $I$ is equal to 0. Suppose a nonzero element $x \in R$ is believed to belong to $I$. Determine whether $x$ can actually exist in $I$. Justify your conclusion.

**Answer Derivation Process**: We are told that $I$ is an ideal in a commutative ring $R$ with no zero divisors, and that every product of two elements of $I$ equals 0. That means $I^2 = \{\sum a_i b_i \mid a_i, b_i \in I\} = (0)$.
Now, assume for contradiction that $I$ contains a nonzero element $x$. Because $x \in I$, we can form the product $x \cdot x = x^2$, which must lie in $I^2$. But $I^2 = (0)$, so $x^2 = 0$.
This implies $x$ would be a nonzero element whose square is 0, making $x$ a zero divisor. However, the ring $R$ is specified to have no zero divisors, so this is impossible.
Therefore, the assumption that $x$ exists nonzero in $I$ is false. Hence, the only possibility is that $I = (0)$.

**Generated Answer**: $I = (0)$.

**Difficulty**:
5.5
**Domain**:
['Algebra', 'Abstract Algebra', 'Ring Theory'], ['Algebra', 'Abstract Algebra', 'Ideals']

## C  REASONING PROCESS UNDER DIFFERENT SYNTHETIC DATA

To compare the characteristics and quality of our synthetic data with those synthetic data from other environments, we show the reasoning processes of GPT5-Mini on these synthetic data as following cases.

In the Synlogic case, it is a typical lies logic reasoning problem. The question is long for describing the problem scenario. However, the GPT5-Mini complete the solution using only 757 tokens. The most of reasoning behaviors in the solutions are hypothetical deductions and without any explicit use of mathematical knowledge or skills. In the Absolute-Zero case, the task is a solving abduction task, where the model is provided with a program and its output and is required to predict the input. GPT5-Mini complete this question using only 638 tokens, employing reverse reasoning to arrive at the final answer. Here, the reasoning is primarily grounded in the logical structure of the code and proceeds through step-by-step execution-based inference. The same as logic games, it involves less math reasoning skills.

**Data Source: Ours**

**Question**:
Let A be a positive real number. Find all values of A for which the inequality

$$16(A+2)^4(2A+1) < 243(A+1)^4.$$

holds.

**Reasoning Process of GPT5-Mini**:
Let's solve the inequality carefully, **step by step**. The inequality is:

$$16(A+2)^4(2A+1) < 243(A+1)^4$$

where $A > 0$.
Step 1: Rewrite the inequality in a nicer form
Divide both sides by $(A+1)^4 > 0$:

$$16(A+2)^4 \frac{2A+1}{(A+1)^4} < 243, \quad 16(2A+1)\left(\frac{A+2}{A+1}\right)^4 < 243$$

Step 2: Simplify the constants
Divide both sides by 16:

$$(2A+1)\left(\frac{A+2}{A+1}\right)^4 < \frac{243}{16}$$

Notice that $243 = 3^5$ and $16 = 2^4$, so:

$$\frac{243}{16} = \frac{3^5}{2^4} = \frac{243}{16}$$

We'll leave it as $\frac{243}{16}$ for now.
Step 3: Make a substitution
Set $x = \frac{A+2}{A+1} > 1$ since $A > 0$. Then:

$$A+2 = x(A+1) \implies A+2 = xA+x$$

$$2 = xA - A + x \implies 2 = A(x-1) + x \implies A(x-1) = 2-x \implies A = \frac{2-x}{x-1}$$

Also,

$$2A+1 = 2 \cdot \frac{2-x}{x-1} + 1 = \frac{3-x}{x-1}$$

So the inequality becomes:

$$(2A+1)\left(\frac{A+2}{A+1}\right)^4 = \frac{3-x}{x-1} \cdot x^4 < \frac{243}{16}$$

$$x^4 \frac{3-x}{x-1} < \frac{243}{16}$$

Step 4: Solve the inequality for $x$
Multiply both sides by $x-1 > 0$:

$$x^4(3-x) < \frac{243}{16}(x-1), \quad 16x^4(3-x) < 243(x-1), \quad 16x^4(3-x) - 243(x-1) < 0, \quad 16x^4(3-x) - 243x + 243 < 0$$

$$-16x^5 + 48x^4 - 243x + 243 < 0$$

Multiply both sides by $-1$ (reverse inequality):

$$16x^5 - 48x^4 + 243x - 243 > 0, \quad 16x^5 - 48x^4 + 243(x-1) > 0$$

Step 5: Factor if possible
Try factoring $16x^5 - 48x^4 + 243(x-1) = 16x^5 - 48x^4 + 243x - 243$. Group:

$$(16x^5 - 48x^4) + (243x - 243) = 16x^4(x-3) + 243(x-1)$$

So the inequality becomes:

$$16x^4(x-3) + 243(x-1) > 0$$

Since $x > 1$, $x-1 > 0$. Also $x-3$ can be positive or negative. Check for equality to find boundary:

$$16x^4(x-3) + 243(x-1) = 0, \quad 16x^4(x-3) = -243(x-1), \quad 16x^4(3-x) = 243(x-1)$$

This is exactly the original equation in Step 4.
Step 6: Solve numerically
We need $1 < x < 3$ because $x > 3 \implies 3-x < 0$ and LHS negative.
Test some values: $x = 1$: LHS $= 16 \cdot 1^4(3-1) = 32$, RHS $= 243 \cdot 0 = 0 \to$ inequality satisfied? $32 < 243(0)$? Not really. Let's check original inequality: better to solve directly. Better approach: the inequality originally:

$$16(A+2)^4(2A+1) < 243(A+1)^4$$

Try rational guess: maybe solution is of the form $A = 1/2$ or 1. $A = 1$: LHS $= 16 \cdot 3^4 \cdot 3 = 3888$, RHS $= 243 \cdot 2^4 = 3888 \to$ equality! $A < 1$: $A = 1/2$: LHS $\approx 1250$, RHS $\approx 1230 \to$ inequality false. $A > 1$: $A = 2$: LHS $= 20480$, RHS $= 19683 \to$ inequality false. $A$ very small, e.g. $A = 0.01$: LHS $\approx 264.9$, RHS $\approx 253 \to$ inequality false
So inequality never holds for $A > 0$, equality at $A = 1$. Conclusion: The inequality **has no positive solutions**. The **only point of equality** is A=1 **Answer:**
No positive A satisfies the strict inequality. Equality holds at A = 1.**(1,899 Tokens)**

**Data Source: Synlogic-Web of Lies**

**Question**:
Solve the following problem step by step. First, think about the reasoning process in the mind and then provide the answer. The reasoning process is enclosed within and the final answer is enclosed within <answer> </answer> tags, respectively, i.e., <think> reasoning process here </think> <answer> answer here</answer>.
In this puzzle, we have 8 people, and each person either always tells the truth or always lies. Given information: the person at the hospital says Matthew tells the truth. Alexander is at the botanical garden. Emma is at the restaurant. the person at the theater says Matthew tells the truth. the person at the hospital says the person at the airport lies. the person at the theater says the person at the airport lies. Mia is at the theater. Mia says Sofia tells the truth. Ava says Matthew lies. the person at the train station mentions that the person at the theater is smart. Matthew is at the train station. the person at the theater says the person at the airport is a good person. the person at the night market says the person at the hospital tells the truth. the person at the train station says Emma tells the truth. the person at the restaurant says Mia tells the truth. the person at the hospital mentions that Matthew is smart. the person at the restaurant discovered errors in their colleague's work report. Alexander says the person at the airport lies. Sofia says the person at the botanical garden tells the truth. Ava is at the airport. the person at the supermarket says the person at the restaurant tells the truth. Ethan is at the night market. the person at the restaurant says Ethan tells the truth. the person at the theater says the person at the restaurant tells the truth. Sophia is at the hospital. the person at the hospital never lies. Sofia is at the supermarket. Please answer: Does the person at the restaurant tell the truth? Does the person at the theater tell the truth? Does the person at the supermarket tell the truth? Does the person at the night market tell the truth? Analyze step by step, and end your response in the last line using this format: The answer is ...

**Reasoning Process of GPT5-Mini**:
Let's carefully reason through this step by step.
<think> We have 8 people at different locations: hospital, botanical garden, restaurant, theater, airport, train station, night market, and supermarket. Each person either always tells the truth or always lies. Some people's identities are given, and some statements are given about others.
**Step 1: Identify known locations and people**
* Sophia → hospital * Alexander → botanical garden * Emma → restaurant * Mia → theater * Ava → airport * Matthew → train station * Ethan → night market * Sofia → supermarket
**Step 2: Use the hospital as a reference**
* Sophia is at the hospital and "never lies," so all statements from Sophia are true. * The hospital person says Matthew tells the truth → Matthew tells the truth. * The hospital person says the airport person lies → Ava lies. * The hospital person says Matthew is smart → not strictly relevant to truth/lie but consistent with Matthew telling the truth.
**Step 3: Use the theater statements**
* Mia is at the theater. Statements from the theater:
* Matthew tells the truth → consistent with Step 2. * Airport person lies → consistent with Step 2 (Ava lies). * Airport person is a good person → this is subjective; we focus on truth/lie. * Restaurant person tells the truth → we'll check consistency with other statements.
**Step 4: Train station statements (Matthew)**
* Matthew at train station:
* Theater person is smart → subjective. * Emma tells the truth → does restaurant tell the truth? We'll see.
**Step 5: Restaurant (Emma)**
* Emma says:
* Mia tells the truth → check consistency. * Ethan tells the truth → check consistency. * Discovered errors in a colleague's report → might imply Emma is careful/truthful but not strictly logical.
**Step 6: Supermarket (Sofia)**
* Supermarket person (Sofia) says: restaurant person tells the truth → this is about Emma.
**Step 7: Night market (Ethan)**
* Night market person (Ethan) says: hospital person tells the truth → Sophia never lies, so Ethan tells the truth (since statement is true).
**Step 8: Analyze statements about restaurant truthfulness**
* Theater person says restaurant tells the truth → Mia says Emma tells the truth? Actually, theater person statement: restaurant person tells the truth → matches Emma at restaurant. * Supermarket person (Sofia) says restaurant tells the truth → both agree → consistent → Emma tells the truth.
**Step 9: Theater truthfulness**
* Mia is at theater. What does Mia say?
* Sofia tells the truth → Sofia at supermarket, does she tell the truth?
* Sofia's statement: restaurant tells the truth → restaurant (Emma) tells the truth → statement is true → Sofia tells the truth → Mia's statement true → Mia tells the truth.
**Step 10: Supermarket truthfulness**
* Sofia at supermarket says restaurant tells the truth → true → Sofia tells the truth.
**Step 11: Night market truthfulness**
* Ethan at night market says hospital person tells the truth → hospital person (Sophia) tells the truth → Ethan tells the truth.
So we conclude:
* Restaurant (Emma) → tells the truth * Theater (Mia) → tells the truth * Supermarket (Sofia) → tells the truth * Night market (Ethan) → tells the truth
All statements are consistent.
</think> <answer> yes, yes, yes, yes </answer> **(757 Tokens)**

---

**Data Source: AbsoluteZero-Solve Abduction Task**

**Question**:
Here is the function f and the output: '1', please predict the input.

```
def f(numbers: list[int], target: int) -> int:
    accumulated_values = []
    current_sum = 0
    for number in numbers:
        current_sum += number
        accumulated_values.append(current_sum)
    result = 0
    for i in range(len(accumulated_values)):
        for j in range(i + 1, len(accumulated_values)):
            if accumulated_values[j] \
                - accumulated_values[i] == target:
                result = accumulated_values[i]
                return result

    return result
```

**Reasoning Process of GPT5-Mini**:
Let's carefully analyze the function `f` and the output '1' to predict the input. We'll go **step by step**.
—

Function Analysis

```
def f(numbers: list[int], target: int) -> int:
    accumulated_values = []
    current_sum = 0
    for number in numbers:
        current_sum += number
        accumulated_values.append(current_sum)
```

* This loop computes **prefix sums** of the list `numbers`. For example, if `numbers = [a, b, c]`, then `accumulated_values = [a, a+b, a+b+c]`.

```
    result = 0
    for i in range(len(accumulated_values)):
        for j in range(i + 1, len(accumulated_values)):
            if accumulated_values[j] - accumulated_values[i] == target:
                result = accumulated_values[i]
                return result
```

* This double loop checks all pairs `(i, j)` such that `i < j`. * If `accumulated_values[j] - accumulated_values[i] == target`, it returns `accumulated_values[i]`.
**Important detail:** the function **returns the first prefix sum `accumulated_values[i]`** that satisfies the condition.
—

```
    return result
```

* If no pair is found, it returns 0.
Given Information
* Output of the function is 1. * So there exists `i < j` such that:

$$accumulated\_values[j] - accumulated\_values[i] = target$$

and

$$accumulated\_values[i] = 1$$

—
Reverse Engineering
Let's denote the prefix sums as $[s_0, s_1, s_2, \ldots]$.
* We know `s_i = 1`. * And `s_j - s_i = target` → `s_j - 1 = target` → `s_j = target + 1`.
The simplest case is a **short list**. Let's try `numbers = [1, something]`:
1. Prefix sums: $[1, 1 + something]$. 2. $i = 0, s\_i = 1$. 3. $s\_j - s\_i = (1 + something) - 1 = something = target$.
Works perfectly. So the minimal input example is:

```
numbers = [1, target]
```

* Then `accumulated_values = [1, 1 + target]`. * $s\_j - s\_i = (1 + target) - 1 = target$. * The function returns $s\_i = 1$.
—
Predicted Input
* `numbers = [1, target]` * `target = target` (same as the function's target argument)
This is the **simplest valid input** that would produce output 1. **(638 Tokens)**

# D PROMPT LIST

## D.1 LLM HYPOTHESIS REJECTION PROMPT

```
You are an assistant helping to validate Lean theorems that have been
compiled successfully, but may contain invalid assumptions.

## Background:
In Lean, theorems with contradictory or impossible assumptions can still
compile successfully. This is because Lean treats such assumptions as
false, which means the conclusion is vacuously true. For example, if a
theorem assumes that a continuous function exists and satisfies a
condition that can never be true, the proof still passes in Lean, even
though the assumptions are flawed.

To detect such vacuous theorems, we convert them to natural language
using the "informalize" tool. Your job is to examine the **assumptions
only**.

## Task:
Given a natural-language version of a Lean theorem (produced by
informalize), check **only whether the assumptions are mathematically
valid and non-contradictory**.

You do **not** need to verify the correctness of the conclusion. Focus
only on whether the assumptions:
1. Contain contradictions
2. Claim the existence of impossible or undefined objects
3. Rely on unrealistic constructions that make the theorem vacuously true

## Output format:
Respond with a structured JSON block like this:

{
  "valid": true or false,
  "reason": "Brief explanation of whether and why the assumptions are
  valid or not."
}

Now analyze the following theorem:
{}
```

## D.2 DIFFICULTY RATING PROMPT AND DISTRIBUTION OF STATEMENTS LEVEL

```
# CONTEXT #
I am a teacher, and I have some high-level olympiad math problems.
I want to categorize the domain of these math problems.

# OBJECTIVE #
A. Summarize the math problem in a brief sentence, describing the
concepts involved in the math problem.
B. Categorize the math problem into specific mathematical domains. Please
 provide a classification chain, for example: Applied Mathematics ->
Probability -> Combinations.
The bibliography is a basic classification framework in the field of
mathematics.
<math domains>
(Too long to show, domain tree here..)
</math domains>

# STYLE #
Data report.

# TONE #
Professional, scientific.

# AUDIENCE #
```

```
Students. Enable them to better understand the domain of the problems.

# RESPONSE MARKDOWN REPORT #
## Summarization
[Summarize the math problem in a brief paragraph.]
## Math domains
[Categorize the math problem into specific mathematical domains,
including major domains and subdomains.]

# ATTENTION #
- The math problem can be categorized into multiple domains, but no more
than three. Separate the classification chains with semicolons(;).
- Your classification MUST fall under one of the aforementioned subfields
; if it really does not fit, please add "Other" to the corresponding
branch. For example: Algebra -> Intermediate Algebra -> Other. Only the
LAST NODE is allowed to be "Other", the preceding nodes must strictly
conform to the existing framework.
- The math domain must conform to a format of classification chain, like
"Applied Mathematics -> Probability -> Combinations".
- Add "=== report over ===" at the end of the report.

<example math problem>
(Too long to show)
</example math problem>

## Summarization
The problem requires finding a value that makes the equation $\frac{1}{9} + \frac{1}{18} = \frac{1}{\square}$.
This involves adding two fractions and determining the equivalent
fraction.

## Math domains
Mathematics -> Algebra -> Prealgebra -> Fractions;

=== report over ===

<example math problem>
(Too long to show)
</example math problem>

## Summarization
The problem asks for the possible values of $n$ for a regular n-sided
polygon that can be completely triangulated into isosceles triangles
using non-intersecting diagonals. The solution involves analyzing the
properties of the diagonals forming isosceles triangles and deducing that
 $n$ can be expressed in terms of powers of 2.

=== report over ===

<math problem>
{Question:{Question Here}}
{Solution:{Solution Here}}
{Source:{Source Here}}
</math problem>
```

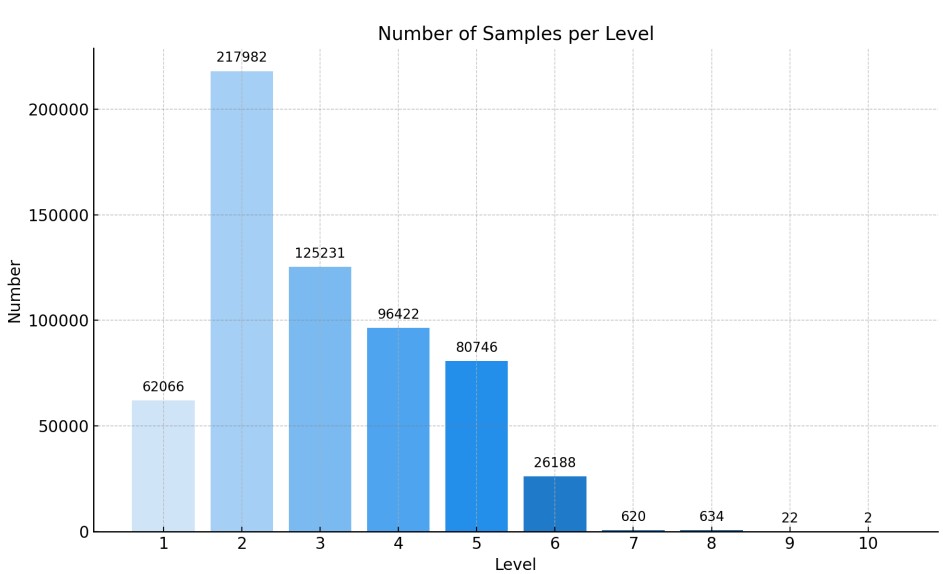

Figure 8: The distribution of collected statements level.

### D.3 PROBLEM GENERATION PROMPT

```
I have a theorem that has already been proven. Please design an **
interesting mathematical solving problem** based on given theorem, where
the theorem serves as the underlying logic for solving it. The
requirements are:
1. The generated problem statement **should not reveal the theorem itself
** (hiding the theorem, so do not mention "proof ... then solve" or
similar terms in the generated problem), and successfully solving this
problem must rely on finding and proving this theorem.
2. **When generating problems involving inequality expressions** (e.g.,
interpret these phrases as **bounds/constraints, not exact extrema**. For
 example, from x ≤ 5 we only know that 5 is an upper bound of x, not that x
 actually attains the maximum value 5. The fact that such a bound is
accepted by the Lean compiler does not imply it represents the maximum or
 minimum value.
3. Only generate problems with a single, fully determined final answer.
The answer must be uniquely specified (e.g. exact value, complete
interval, finite set, Yes/No). Avoid problems where the solution could be
 partial or admit multiple valid answers.
4. You can **analysis and generate its solution directly using the proven
 theorem to check the answer correctness**.
5. The final answer is the summary of reasoning process, **it can be a
numerical value, a range, a short phrase or a Boolean. ** It should be
presented in the format: Final Answer: {}.
6. Finally, the output must **follow the exact structure (structured
output)**: Generated Question: {}
Solution: {}
Final Answer: {}.
##Example1
Given Theorem: Suppose a, b, and c are real numbers such that the sum of
their squares is 3. Then the sum of their sixth powers is at least the
cube of the sum of their pairwise products divided by 9 and also divided
by √3+2.
The Output Result:
Generated Question: Let a be a positive real number. Does there exist an
integer S such that 240 < S < 250 and the inequality \frac{16(a+2)^4(2a+1)
}{(a+1)^4} > S holds for some value of a? Answer with Yes or No.
Solution: The given theorem can be rewritten as 16(a+2)^4(2a+1)/(a+1)^4 ≥ 243 for all
a > 0. Let f(a) denote the expression on the left, the theorem states a
bound of f(a) is 243. Although the theorem itself does not state if this
bound is the actual minimum value, we can still answer the question with
Yes.
Final Answer: Yes, such an integer S exists.

##Example2
Given Theorem: Suppose you have three real numbers \( x \), \( y \), and
\( z \) such that their sum is zero, and the sum of their pairwise
products is \(-3\). Then the expression \( x^3 y + y^3 z + z^3 x \)
equals \(-9\).
The Output Result:
Generated Question: Let \( a, b, c \) be real numbers such that \( a + b
+ c = 0 \) and \( ab + bc + ca = -3 \). Calculate the value of the
expression \( 3 \times (a^3 b + b^3 c + c^3 a) \)?
Solution: According to the theorem, (a^3 b + b^3 c + c^3 a) is \(-9\).
This means that \( 3 \times (a^3 b + b^3 c + c^3 a) \) is 3 * (-9) = -27.
Final Answer: \(-27\).

##Example3
Given Theorem: For any positive real number a, the expression 16 \times (
a + 2)^4 \times (2a + 1) is greater than or equal to 243 \times (a + 1)
^4.
The Output Result:
```

```
Generated Question: Let a be a positive real number. Find all values of a
 for which the inequality 16 \times (a + 2)^4 \times (2a + 1) < 243 \
times (a + 1)^4  holds.
Solution: Consider the given inequality:16    imes (a + 2)^4  imes (2a +
1) < 243     imes (a + 1)^4. We are to find all positive real numbers a
such that the left side is strictly less than the right side. However, by
 analyzing the structure of the expressions and considering all positive
real values for a, we note that the left side is always greater than or
equal to the right side for any positive real a. This means that there
are no positive real values of a for which the inequality is satisfied.
Final Answer: No such a exists.

Now please design a mathematical solving problem based on following
theorem.
Given Theorem: {}
The Output Result:
```

## D.4 REASONING BEHAVIOR EXTRACTION PROMPT

```
Below is a chain-of-reasoning generated by a Language Model when
attempting to solve a
math problem. Evaluate this chain-of-reasoning to determine whether it
demonstrates
beneficial problem-solving behaviors that deviate from typical linear,
monotonic reasoning
patterns commonly observed in language models.
<start_of_reasoning>
{input}
<end_of_reasoning>
Specifically, actively identify and emphasize beneficial behaviors such
as:
(1) Backtracking: Explicitly revising approaches upon identifying errors
or dead ends (e.g., "This
approach won't work because...").
(2) Verification: Systematically checking intermediate results or
reasoning steps (e.g., "Let's
verify this result by...").
(3) Subgoal Setting: Breaking down complex problems into smaller,
manageable steps (e.g.,
"To solve this, we first need to...").
(4) Enumeration: Solving problems by exhaustively considering multiple
cases or possibilities.
Additionally, remain attentive to and encourage the identification of
other beneficial behaviors
not explicitly listed here, such as creative analogies, abstraction to
simpler cases, or insightful
generalizations.
Important:
Clearly specify each beneficial behavior you identify.
Provide explicit examples from the reasoning chain.
If no beneficial behaviors are observed, explicitly return an empty list.
Provide your evaluation clearly, formatted as follows:
```json
{
 "behaviour": "",
 "example": ""
}
```
```

