# OpenReview forum: "F1-Reasoner: Synthesizing Verifiable Reasoning Data From Formal Math Statements"
_ICLR.cc/2026/Conference — Submitted to ICLR 2026_

### Official Review · Reviewer_JMzx · 2025-10-22

**Soundness:** 2
**Presentation:** 2
**Contribution:** 2
**Rating:** 6
**Confidence:** 2

**Summary:**

The paper introduces a framework for synthesizing verifiable mathematical reasoning data from formal theorem proving systems. The framework is a pipeline consisting of 4 steps, where theorems are gathered, filtered, transformed into different problems, and used for training a model. The paper evaluates the approach on 6 benchmarks and compares with multiple alternative methods.

**Note:** *I am not very familiar with this field, so I do not know the existing state-of-the-art. This also means I cannot evaluate well how novel the approach is compared to existing methods, nor any claims about this.*

**Strengths:**

- Thorough experimental evaluation of multiple competitors on 6 benchmarks. Additional experiments investigate "informal theorem proving".
- The results show that the approach generally beats the competitors.

**Weaknesses:**

- For multiple components of the proposed method, it is unclear why they are 'useful', i.e. no real arguments are given as to what they 'bring' to the method (see Questions 3 and 6). Ablation studies would also help with this.
- It seems an unfair advantage that the approach trains on one of the benchmarks (MATH) while the other approaches seem to not do this?
- There are many typos and language mistakes to the point that it is sometimes hard to understand what is meant (see below)

**Typos and mistakes:**
- line 42: leverage[s]
- line 46: "has" -> "have"
- line 51: "which approach" -> ". This approach"
- line 74: "potentiality" -> "possibility"
- line 86: obtain [the] F1-Reasoner model
- line 99: generalize[s]
- line 100: a[n]
- line 106: proven by [a] formal theorem proving system
- line 162: "the high difficulty problem is" -> "high difficulty problems are"
- line 183: by [the] community
- line 200: question[s]
- line 200: answer[s]
- line 256: "the verified model" -> "a verified model"
- line 297: "()" (I don't know what is missing here)
- line 303: "use combining" -> "combine" or "use"
- line 451: missing space after ")"
- line 481: a[n]

**Questions:**

1. What do you mean with "inconsistent hypotheses" (Section 2.2)? I presume this means statements where the premise trivially evaluates to false?
2. What does "by sorry" mean (Figure 2)?
3. Why do you employ the LLM-as-judge strategy in addition to the rejection using the theorem proving (line 160)? The text does not explain why the theorem proving component is insufficient.
4. You only keep statements with difficulty > 5 (line 172). Did you evaluate whether instead keeping these statements would improve the model's accuracy on statements with difficulty < 5?
5. How sensitive are results to the difficulty filtering threshold (> 5)?
6. What is "the provided scorer" (line 178)? Is this part of the Deita selection pipeline? Could you clarify this in the text?
7. Why is the decontamination step useful (line 182)? What does data contamination imply? This is not explained in the text.
8. Why are some competitors only used with some base models (Table 1)?
9. I do not understand this sentence (lines 373-374): "These approaches can generate correct but intermediate problems, their correctness is hard to guarantee as difficulty increases." Do you mean the generated problems of such approaches are too hard to properly evaluate?

---

> ### Author Response · Authors · 2025-11-26
> **Response to Reviewer JMzx (1/2)**
>
> Thanks for your time and valuable comments. We sincerely appreciate your feedback on F1-Reasoner. We now present a detailed response to address your concerns and comments:
>
> > W1: For multiple components of the proposed method, it is unclear why they are 'useful', i.e. no real arguments are given as to what they 'bring' to the method (see Questions 3 and 6).
> Q3: Why do you employ the LLM-as-judge strategy in addition to the rejection using the theorem proving (line 160)?
> Q6: What is "the provided scorer" (line 178)? Is this part of the Deita selection pipeline?
>
> We apologize for the confusion in our quality control pipeline. Our filtering strategies build upon techniques that have shown effectiveness in many prior works and are increasingly used as useful methods for improving data quality. Below, we clarify two specific components: hypothesis rejection and the Deita selection filtering pipeline.
>
> (1) **Hypothesis Rejection Strategy**
>
> The hypothesis rejection relies on formal provers to prove the premises of the given statement are false. If Lean successfully verifies the generated proof, we conclude that  the premises of the original statement are inherently contradictory. This task is quite simple for provers as it only requires checking whether the assumptions lead to a contradiction. Nonetheless, long-tail inconsistencies may still be missed due to prover generalization, so we add an extra LLM-as-judge filter. This double-checking mechanism reduces the risk of incorporating inconsistent statements.
>
> (2) **Deita Selection Pipeline**
>
> Yes, the “provided scorer” refers to the LLM-based scoring model released by the Deita selection pipeline. We directly use their trained scorer to evaluate complexity. We have revised the paper to explicitly clarify that this scorer originates from Deita to avoid confusion, thanks for your advice.
>
> > W2: It seems an unfair advantage that the approach trains on one of the benchmarks (MATH) while the other approaches seem to not do this?
>
> Thank you for raising this concern. Our experiments involve two distinct types of comparisons, serving different purposes:
>
> (1) **F1 vs. other synthetic-environment datasets (e.g., Absolute-Zero, SynLogic)**
>
>  These comparisons aim to evaluate which environment produces higher-quality synthetic data. In these settings, our dataset is significantly smaller (e.g., SynLogic has 49k examples), yet our data achieves better performance with fewer samples, suggesting higher data efficiency.
>
> (2) **F1-Mix vs. MATH**
>
>  Our Mix dataset is constructed by adding our F1 data in MATH, with the goal of **evaluating whether our synthetic data provides incremental benefits when combined with existing human-curated data.** The results show that Mix consistently improves performance, and the gains are more pronounced for larger models. On Qwen3-8B, F1-Mix even surpasses General-Reasoner-14B. **Following your suggestion, we evaluate whether other synthetic datasets yield similar gains when mixed with MATH. We conducted a new experiment ("SynLogic-Mix") on Qwen3-8B**:
>
> | **Model Name** | **AVG.** | **AMC** | **Minerva** | **MATH** | **Olympiad** | **AIME25** | **AIME24** |
> |---|---|---|---|---|---|---|---|
> | SynLogic-Mix | 49.65 | 63.80 | 60.70 | 85.20 | 51.70 | 16.80 | 20.10 |
> | F1-Reasoner-Mix | **54.42** | **72.30** | 66.90 | **87.60** | **55.60** | **20.40** | **25.10** |
>
>
> F1-Reasoner-Mix outperforms SynLogic-Mix by 4.77 points on average, indicating that our data is more helpful when used as additional training data compared to data generated from SynLogic.
>
>
> > W3: There are many typos and language mistakes to the point that it is sometimes hard to understand what is meant
>
>
> Thank you for pointing out these issues. We apologize for the oversight. **We have addressed all of these issues including grammatical errors, missing references and unclear wording.** Please kindly refer to the revision.
>
>
>
> > Q1: What do you mean with "inconsistent hypotheses" (Section 2.2)? I presume this means statements where the premise trivially evaluates to false?
>
> Yes, "inconsistent hypotheses"  means the assumptions within the statement conflict with each other. For example, in Figure 2, substituting (x = a) into the definition formula gives (f(a) = 1^2 + 1 + 1 = 3), while the problem simultaneously imposes the constraint (f(a) = 0). Since (3 \neq 0), the same function cannot have two different values at the same point, creating a logical contradiction. Such a conjecture is therefore meaningless, and including it would negatively affect the quality of the dataset.
>
> > Q2: What does "by sorry" mean?
>
>
> We apologize for the confusion. In Lean, “by sorry” indicates a proof that is yet to be completed. In Figure 2, our intention was to show how a statement’s proof goal is transformed into a goal for proving the premises are contradictory. To avoid misleading readers, we have removed "by sorry" from the figure in the revised version.

---

> > ### Author Response · Authors · 2025-11-26
> > **Response to Reviewer JMzx (2/2)**
> >
> > > Q4: You only keep statements with difficulty > 5 (line 172). Did you evaluate whether instead keeping these statements would improve the model's accuracy on statements with difficulty < 5?
> > Q5: How sensitive are results to the difficulty filtering threshold (> 5)?
> >
> > We set the difficulty threshold to ≥ 5 primarily based on prior empirical evidence rather than hyperparameter tuning. It has two main reasons:
> >
> > - In earlier ablation studies on the MATH subset used by simpleRL, which provides AoPS difficulty labels ranging from Level 3 to 5. We trained models separately on Level 3, Level 4, Level 5, and Levels 3–5 combined. Interestingly, we observed that training **only on Level 5 data** resulted in better average performance than training on the union of Levels 3–5, despite using fewer training samples.
> >
> >
> > | Model | AIME24 | AMC23 | MATH500 | MinervaMath | OlympiadBench | Avg. |
> > | :--- | :--- | :--- | :--- | :--- | :--- | :--- |
> > | Qwen2.5-3B-Base | 3.3 | 20.0 | 41.2 | 11.8 | 10.8 | 17.4 |
> > | Qwen2.5-3B-L3 | **10.0** | 27.5 | 57.6 | 21.7 | 26.2 | 28.6 |
> > | Qwen2.5-3B-L4 | **10.0** | 32.5 | 61.2 | **26.5** | 27.6 | 31.6 |
> > | Qwen2.5-3B-L5 | 6.7 | **42.5** | 61.0 | 21.7 | **27.9** | **32.0** |
> > | Qwen2.5-3B-All | 6.7 | 40.0 | **61.6** | 23.5 | 26.5 | 31.7 |
> >
> > This result suggests that simply incorporating more lower-difficulty data does not improve generalization; instead, **higher-difficulty samples lead to better mathematical reasoning capabilities**, likely because they contain richer multi-step reasoning structures.
> >
> > - We then scored the difficulty of our seed data using an LLM, calibrated with the AoPS rating. As shown in Figure 8 (Section D.2), most seed examples fall between Level 1 and Level 4. **Including these easier samples could potentially hurt performance and also significantly increase data generation cost.** Therefore, we only kept seed data with difficulty ≥ Level 5. This choice was based on performance and cost considerations, **rather than tuning thresholds and selecting 5 as a hyperparameter.**
> >
> >
> > > Q7: Why is the decontamination step useful (line 182)? What does data contamination imply? This is not explained in the text.
> >
> > The decontamination step is necessary because part of our data is derived from Leanworkbook, where some statements originate from existing mathematical problems. To avoid potential overlap with benchmark questions, we perform data filtering to remove statements that are similar to testset questions. This ensures that our evaluation is not affected by data leakage and that the reported performance is reliable.
> >
> > > Q8: Why are some competitors only used with some base models (Table 1)?
> >
> >
> >
> > We apologize for the confusion. For some baselines (e.g., Absolute-Zero and SynLogic), **we reported the results directly from their papers, which primarily trained on Qwen2.5.** As a result, the corresponding results on Qwen3 models were missing in our initial submission.
> > To provide a more complete comparison, we use the data from SynLogic ( Absolute-Zero does not release training data) and conducted additional experiments on **Qwen3-8B**. The results are shown below:
> >
> > | **Model Name** | **AVG.** | **AMC** | **Minerva** | **MATH** | **Olympiad** | **AIME25** | **AIME24** |
> > |---|---|---|---|---|---|---|---|
> > | SynLogic | 44.72 | 55.00 |60.30 | 80.00 | 46.40 | **13.90** | 12.70 |
> > | F1-Reasoner | **46.20** | **60.80** | **59.60** | **80.60** | **40.70** | 12.20 | **20.00** |
> >
> >
> > We observe that on Qwen3-8B, our F1-Reasoner outperforms SynLogic by 1.48 points on average. **This consistent superiority demonstrates that our generated data remains effective even on stronger backbones.**
> >
> >
> >
> > > Q9: I do not understand this sentence (lines 373-374): "These approaches can generate correct but intermediate problems, their correctness is hard to guarantee as difficulty increases." Do you mean the generated problems of such approaches are too hard to properly evaluate?
> >
> > We compare the methods that generate both the problem statement and its answer solely via an LLM, without validation from a formal system. **Such approaches may produce correct problems when the difficulty remains  at a medium level** (i.e., within the model’s reasoning capability). **However, when generating hard problems, the model often produces incorrect answers, as these problems exceed its reasoning capabilities and there is no formal prover to guarantee correctness.** These errors can accumulate and negatively impact downstream training (as demonstrated in Section 3.3 _Training on Data without Formal System Verification_. of our experiments).
> >
> > > In summary, we thank the reviewer again for highlighting key points of improvement and opportunities for further work. We hope that these clarifications will lead to a more favorable assessment of our submission.

---

> > > ### Comment · Reviewer_JMzx · 2025-11-26
> > >
> > > I thank the authors for addressing my concerns. I will increase my score accordingly.
> > >
> > > Could the authors explain / mention in the text the following answers that were given to my questions?
> > >
> > > > The hypothesis rejection relies on formal provers to prove the premises of the given statement are false. If Lean successfully verifies the generated proof, we conclude that the premises of the original statement are inherently contradictory. This task is quite simple for provers as it only requires checking whether the assumptions lead to a contradiction. Nonetheless, long-tail inconsistencies may still be missed due to prover generalization, so we add an extra LLM-as-judge filter. This double-checking mechanism reduces the risk of incorporating inconsistent statements.
> > >
> > > > In earlier ablation studies on the MATH subset used by simpleRL, which provides AoPS difficulty labels ranging from Level 3 to 5. We trained models separately on Level 3, Level 4, Level 5, and Levels 3–5 combined. Interestingly, we observed that training only on Level 5 data resulted in better average performance than training on the union of Levels 3–5, despite using fewer training samples.
> > >
> > > > We then scored the difficulty of our seed data using an LLM, calibrated with the AoPS rating. As shown in Figure 8 (Section D.2), most seed examples fall between Level 1 and Level 4. Including these easier samples could potentially hurt performance and also significantly increase data generation cost. Therefore, we only kept seed data with difficulty ≥ Level 5. This choice was based on performance and cost considerations, rather than tuning thresholds and selecting 5 as a hyperparameter.
> > >
> > > > The decontamination step is necessary because part of our data is derived from Leanworkbook, where some statements originate from existing mathematical problems. To avoid potential overlap with benchmark questions, we perform data filtering to remove statements that are similar to testset questions. This ensures that our evaluation is not affected by data leakage and that the reported performance is reliable.
> > >
> > > > We reported the results directly from their papers, which primarily trained on Qwen2.5.

---

> > > > ### Author Response · Authors · 2025-11-27
> > > > **Response to Reviewer JMzx**
> > > >
> > > > Dear Reviewers,
> > > >
> > > > Thank you very much for increasing your recommendation for our paper. We sincerely appreciate your thoughtful evaluation. We will incorporate the explanations above into our revision to make the paper clearer for readers. Thank you again for helping us improve the quality of our work!
> > > >
> > > > best wishes,
> > > >
> > > > Authors of Paper 9641

---

### Official Review · Reviewer_7Hrg · 2025-10-26

**Soundness:** 3
**Presentation:** 3
**Contribution:** 2
**Rating:** 6
**Confidence:** 3

**Summary:**

This work proposes a data synthesis framework that converts formal math statements (in Lean) into high-quality, verifiable natural language reasoning data. The authors then use this synthesized data to train a model, which ultimately achieves superior performance compared to existing open-sourced models.

**Strengths:**

The overall quality and clarity of this work are good, presenting its motivation and methodology in detail.

A key distinction of this paper is its novel approach. While many works leverage natural language reasoning data for formal theorem proving, this work explores the alternative direction: synthesizing natural language data from formal data. This method might be very important for the future development of model reasoning.

**Weaknesses:**

The main problem concerns the motivation of this work. Many formal data sets are originally translated from natural language statements. This paper now suggests translating them back into natural language, a process that seems to offer minimal gain.

(Specifically, if the formal data already has a corresponding natural language version, what is the key difference or added value between the newly "translated-back" data and the original natural language data?)

**Questions:**

Can you apply your methodology (e.g., quality control) also on natural language data?

---

> ### Author Response · Authors · 2025-11-26
> **Response to Reviewer 7Hrg (1/1)**
>
> Thank you for reviewing our work. Below, we will address your concerns and suggestions point by point:
>
> > W1: The main problem concerns the motivation of this work. Many formal data sets are originally translated from natural language statements. This paper now suggests translating them back into natural language, a process that seems to offer minimal gain.
> (Specifically, if the formal data already has a corresponding natural language version, what is the key difference or added value between the newly "translated-back" data and the original natural language data?)
>
> We sincerely apologize for the confusion. We would like to clarify the motivation about our work:
>
> - **Motivation**: Our primary goal is to improve the informal mathematical problem-solving ability of large language models. With the success of RLVR, **obtaining reliable QA pairs has become particularly important for reinforcement learning.** Many recent works synthesize data inside artificial environments to construct such correct reasoning QA pairs [1][2]. However, these environments are often far from the real mathematical domain.
> Therefore, we explore **whether it is possible to synthesize QA pairs directly from the formal mathematical environment**, so that the resulting reasoning data are not only genuinely mathematical but also reliable in correctness, which is crucial for RLVR training.
>
>
> - **Source of our statements**: Most of our formal statements come from LLM Conjecturer[3], which **generates new formal conjectures from scratch** rather than translating existing natural-language math questions. These conjectures are then validated by a formal proving system, and only those that pass verification are retained. The resulting verified statements serve as reliable seed data for producing correct QA pairs. **In the final dataset of 19,112 samples, 17,630 are generated from newly synthesized statements, while only 1,482 originate from translated statements.** We believe that ongoing advances of conjecture-generation techniques in the formal math community will yield more diverse and high-quality conjectures.
>
> - **Improving theorem understanding and robustness:**: **Our approach can generate multiple diverse questions from the same theorem.** This encourages the model to engage with the theorem from different perspectives, which improves its understanding of the underlying theorem knowledge and enhances robustness across related problems. Therefore, It also serves as a data augmentation method by generating diverse questions from the same theorem.
>
> We hope this helps clarify the motivation and advantages of our work.
>
> > Q1: Can you apply your methodology (e.g., quality control) also on natural language data?
>
>
>
> Thank you for the question. Some parts of our methodology can indeed be applied to natural language data. Specifically:
>
> 1. **Difficulty, diversity, and data de-noising strategies are all applicable.**
>
> - Difficulty: LLM-as-Judge can be adapted with appropriate prompts to assess the difficulty of natural language problems.
> - Diversity: We use Deita, which is originally designed for natural language problems and can directly encourage diverse question generation.
> - Data decontamination: Computing embeddings of problems works equally well for both formal and informal data, allowing us to filter out redundant examples.
>
>
> 2. **Hypothesis Rejection strategy is specific to formal statements.**
>
> The hypothesis rejection relies on formal provers to prove the premises of the given statement are false. If Lean successfully verifies the generated proof, we conclude that  the premises of the original statement are inherently contradictory. This approach does not directly apply to purely natural language problems, but it is an effective mechanism for identifying invalid formal statements.
>
>
> We hope this explanation clarifies which parts of quality control can generalize to natural language data.
>
> > In summary, we thank the reviewer again for highlighting key points of improvement and opportunities for further work. We hope that these clarifications will lead to a more favorable assessment of our submission.

---

### Official Review · Reviewer_9RnE · 2025-10-27

**Soundness:** 2
**Presentation:** 3
**Contribution:** 2
**Rating:** 4
**Confidence:** 4

**Summary:**

The paper proposes a framework for synthesizing high-quality, verifiable informal reasoning data for training LLMs with formal language.  The paper contributes a 4-stage pipeline that addresses the scarcity of such data, which currently relies on limited manual annotation or artificial environments. In the experiments, the models trained on these synthesized data show strong performance. The authors also study the reasoning behaviour of the models trained with this pipeline.

**Strengths:**

1. The core concept of using a formal proving system as a source for generating informal reasoning problems is excellent.
2. The experimental results are strong.
3. The paper demonstrates the methods clearly.

**Weaknesses:**

1. The choice of provers for the "Statements Collection" phase is questionable. InternLM2.5-StepProver and STP-Prover are weak and outdated. More powerful open-source ones, such as Kimina-Prover, DeepSeek-Prover-v2, were available well before the ICLR 2026 submission deadline. Therefore, this paper is expected to be based on these new models, since the "Statements Collection" phase is the foundation of the proposed pipeline.
2. The methodology introduces biases to the selected data by using formal provers' success as the filter. The paper aims to improve general informal reasoning, but its training data is filtered through the narrow bottleneck of a specific prover's capabilities. This risks training the LLM on a "prover-friendly" subset of mathematics, excluding many correct and difficult problems that the chosen provers simply failed to solve. While the authors introduce methods like difficulty sampling, this only samples from an already-biased pool and does not mitigate this fundamental issue.
3. The generalizability of the "Problem Generation" step is questionable. While the paper provides one concrete example of math inequalities, it is not clear how this method would apply to other, more diverse types of mathematical problems, especially those with numbers or math terms as their answers, which is common in informal math. Actually, the example provided is really weak. The answer to the generated problem is whether correct or wrong, making it easy to guess.
4. The paper presents an interesting analysis in Section 3.3 (Figures 6 and 7), claiming F1-Reasoner exhibits "richer thinking behaviors" and more efficient reasoning. However, the paper doesn't provide an analysis connecting why its specific data generation method causes this outcome. What properties of the formal-to-informal data pipeline encourage these specific behaviors?

**Questions:**

Please answer the questions raised in the Weaknesses part.

---

> ### Author Response · Authors · 2025-11-25
> **Response to Reviewer 9RnE (1/3)**
>
> Thanks for your time and valuable comments. We sincerely appreciate your feedback on F1-Reasoner. We now present a detailed response to address your concerns and comments:
>
> > W1:  The choice of provers for the "Statements Collection" phase is questionable. InternLM2.5-StepProver and STP-Prover are weak and outdated. Therefore, this paper is expected to be based on these new models, since the "Statements Collection" phase is the foundation of the proposed pipeline.
> W2: The methodology introduces biases to the selected data by using formal provers' success as the filter. The paper aims to improve general informal reasoning, but its training data is filtered through the narrow bottleneck of a specific prover's capabilities. This risks training the LLM on a "prover-friendly" subset of mathematics, excluding many correct and difficult problems that the chosen provers simply failed to solve.
>
> Thank you for these thoughtful comments. We would like to clarify that the contribution of our work lies in the F1 dataset and a general and scalable data generation pipeline that converts provable formal statements into challenging informal QA problems.
>
> (1) **The F1 dataset is high-quality and challenging**: Even though our problems originate from statements proven by STP-prover and InternLM2.5-StepProver, **these generated QA pairs are already challenging due to such special problem type: the model must first identify the implicit theorem knowledge, then reason to derive it and finally apply it to solve the question.** As shown in Figure 3, solving such a question requires GPT-5-Mini reasoning in 2000 tokens to arrive at the answer. Beside, **the strong performance improvements on F1-Reasoner further demonstrate that the dataset is effective and difficult.**
>
>
> (2) **Our generation pipeline generalizes to stronger provers**: Our proposed generation pipeline is general and scalable, **as the number of candidate statements increases and stronger provers are incorporated, the pipeline can produce additional high-quality QA pairs for downstream training.** To empirically demonstrate this scalability, we extended the pipeline to incorporate Leanworkbook statements which are proven by **Goedel-Prover**, producing **6.5k** additional QA pairs that form the F1-Plus dataset. The performance on Qwen2.5-7B is shown below:
>
> | **Model Name** | **AVG.** | **AMC** | **Minerva** | **MATH** | **Olympiad** | **AIME25** | **AIME24** |
> |---|---|---|---|---|---|---|---|
> | F1-Reasoner | 38.15 | 48.00 |  **52.90** | **74.40** | 38.10 | 3.80 | 11.70 |
> | F1-Reasoner-Plus | **39.32** | **53.00** | 52.60 | **74.40** | **38.70** | **5.40** | **11.80** |
>
>
> We observe that F1-Reasoner-Plus outperforms F1-Reasoner by 1.17 points on Avg, **yielding consistent improvements across the majority of benchmarks, which verifies the scalability of our data generation framework.**

---

> > ### Author Response · Authors · 2025-11-25
> > **Response to Reviewer 9RnE (2/3)**
> >
> > > W3: The generalizability of the "Problem Generation" step is questionable. While the paper provides one concrete example of math inequalities, it is not clear how this method would apply to other, more diverse types of mathematical problems, especially those with numbers or math terms as their answers, which is common in informal math. Actually, the example provided is really weak. The answer to the generated problem is whether correct or wrong, making it easy to guess.
> >
> >
> > Thank you for the comment. We address the concerns regarding **domain generalizability** and **problem difficulty** below.
> >
> > (1) **Generalizability across mathematical domains**: Yes, our approach can generalize to multiple mathematical domains. To analyze the domain coverage of our dataset, we adopt an LLM-as-Judge tagging method to automatically assign each problem to a domain category. After tagging all samples, **we observe that our dataset spans a wide range of mathematical domains, including algebra, analysis, discrete mathematics, geometry, and number theory.** This demonstrates both the domain diversity of the dataset and the generalizability of our problem generation pipeline.
> >
> > | Main Domain | Included Subcategories |
> > |-------------|-----------------------|
> > | **Algebra** | Algebra, Abstract Algebra, Linear Algebra, Complex Numbers |
> > | **Analysis** | Analysis, Mathematical Analysis, Calculus, Complex Analysis, Trigonometry |
> > | **Discrete Mathematics** | Discrete Mathematics, Combinatorics, Order Theory |
> > | **Geometry & Topology** | Geometry, Topology |
> > | **Number Theory** | Number Theory |
> > | **Logic & Foundations** | Mathematical Logic, Logic, Set Theory, Mathematical Foundations, Foundations of Mathematics |
> > | **Applied Mathematics** | Applied Mathematics, Computer Science, Mathematical Sciences |
> > | **Other** | Mathematics, Pure Mathematics, Other |
> >
> > (2) **Problem difficulty and diversity of answer types**: To encourage question diversity, **we instruct the model to generate problems with various answer formats, including Boolean, numeric, and string expressions,etc.** As a result, our dataset contains a large number of non-trivial questions across diverse mathematical domains and answer types beyond binary predicting. **To facilitate understanding, we provide the following examples to illustrate how it works in other math domains:**
> >
> > | **Statement** | **Question** | **Answer** |
> > |---------------|--------------|------------|
> > | Suppose $x$, $y$, and $z$ are real numbers satisfying the following equations: $x - 5 = -(z - 5)(3z^2 + 15z + 19)$, $y - 5 = -(x - 5)(2x^2 + 10x + 13)$, $z - 5 = -(y - 5)(y^2 + 5y + 7)$, and also that $x$, $y$, and $z$ are all non-negative. Then it must be the case that $x = 5$, $y = 5$, and $z = 5$. | Three friends, Alice, Bob, and Carol, are comparing three numbers $a$, $b$, and $c$, each representing the amount of water (in liters) in their individual water tanks. The relationship between their amounts satisfies: $a - 5 = -(c - 5)(3c^2 + 15c + 19)$, $b - 5 = -(a - 5)(2a^2 + 10a + 13)$, $c - 5 = -(b - 5)(b^2 + 5b + 7)$. It is also known that none of the tanks contain a negative amount of water. Determine the amount of water in each tank. | a = 5 liters, b = 5 liters, c = 5 liters |
> > | Suppose $f$ is a strictly decreasing function from the real numbers to the real numbers, and it satisfies the property that for all real numbers $x$ and $y$, $f(f(x) + y) = f(y - x) - f(0)$. Then, for all real numbers $x$, $f(x) = -x + f(0)$. | Let $g:\mathbb{R}\to \mathbb{R}$ be a strictly decreasing function such that for some constant $c$, the following relation is satisfied for all real numbers $u$ and $v$: $g(g(u)+v) = g(v-u) - c$. Suppose further that $g(2)=3$. What is the value of $g(10)$? | -5 |
> >
> > We provide additional examples in **Appendix B**, please check.

---

> > > ### Author Response · Authors · 2025-11-25
> > > **Response to Reviewer 9RnE (3/3)**
> > >
> > > > W4: The paper presents an interesting analysis in Section 3.3 (Figures 6 and 7), claiming F1-Reasoner exhibits "richer thinking behaviors" and more efficient reasoning. However, the paper doesn't provide an analysis connecting why its specific data generation method causes this outcome. What properties of the formal-to-informal data pipeline encourage these specific behaviors?
> > >
> > > Thank you for raising this concern. Yes, we analyzed the solutions of F1-Reasoner and observed that it exhibits richer reasoning behaviors and solves problems with fewer tokens. We believe this emerges from the nature of our generated data: many of our problems require various reasoning strategies (i.e, enumeration, backtracking, subgoal planning) to discover implicit theorem knowledge and solve them, as illustrated by the case in Figure 3. **Therefore, such problems can activate diverse reasoning behaviors during training, which enables the model to use a richer set of thinking behaviors during inference.**  Moreover, by applying these behaviors and observing which strategies succeed, **the model learns to associate specific strategies with problem types.** It can select better strategies during inference, leading to more efficient reasoning.
> > >
> > > **A similar phenomenon is observed in [1], where introducing more cognition thinking data in continual pretraining led to more diverse reasoning behaviors.** The key difference is that they inject reasoning behavior data in the pretraining stage, whereas in our method such behaviors emerge during the RL stage. We will revise the paper to clarify this connection and include discussion reflecting your suggestion. Thank you for the insightful comment.
> > >
> > > [1] Cognitive behaviors that enable self-improving reasoners, or, four habits of highly effective stars (COLM, 2025)
> > >
> > > > In summary, we thank the reviewer again for highlighting key points of improvement and opportunities for further work. We hope that these clarifications will lead to a more favorable assessment of our submission.

---

> ### Comment · Reviewer_9RnE · 2025-11-26
>
> I have read the authors' reply and increased the Soundness and Contribution scores.

---

> > ### Author Response · Authors · 2025-11-26
> > **Response to Reviewer 9RnE**
> >
> > Dear Reviewer,
> >
> > Thank you very much for your recognition of our response and for increasing the Soundness and Contribution scores. We truly appreciate the time you spent reviewing our work.
> >
> > If possible, we would be grateful if you could also consider strengthening your overall recommendation. If you still have any concerns or questions, we would be more than happy to provide further clarification.
> >
> > Thank you again for your careful review and feedback.
> >
> > best wishes,
> >
> > Authors of Paper 9641

---

### Official Review · Reviewer_vBxm · 2025-10-30

**Soundness:** 3
**Presentation:** 2
**Contribution:** 3
**Rating:** 4
**Confidence:** 4

**Summary:**

This paper introduces F1-Reasoner, a framework for synthesizing verifiable reasoning data from formal mathematical statements to train large reasoning models (LRMs). The approach consists of four main components: (1) Statement Collection from formal theorem proving systems, (2) Quality Control through hypothesis rejection, difficulty assessment, and diversity sampling, (3) Problem Generation that converts formal statements into question-answer pairs, and (4) Model Training using reinforcement learning with verifiable rewards (RLVR). The authors synthesize 19k+ high-quality mathematical problems at levels 5-10 and demonstrate that F1-Reasoner consistently outperforms baselines across six challenging mathematical reasoning benchmarks, including models trained on synthetic data from other environments like SynLogic and Absolute-Zero.

**Strengths:**

1. The use of formal theorem proving systems to ensure correctness of synthetic reasoning data is innovative in this field.
2. The four-stage framework (Statement Collection, Quality Control, Problem Generation, Model Training) is well-designed and addresses multiple aspects of data quality.
3. Consistent improvements across multiple base models (Qwen2.5-7B, Qwen3-4B/8B) and six challenging benchmarks demonstrate the effectiveness of the approach.
4. The paper shows that F1-Reasoner generalizes to informal theorem proving and exhibits richer reasoning behaviors, providing additional validation.

**Weaknesses:**

1. Limited experimental scale: Experiments are conducted only on models up to 8B parameters, while some baselines use larger models (14B, 32B). This limits the conclusions about the approach's effectiveness at scale.

2. Unclear problem generation process: The conversion from formal statements to natural language problems could be better explained. The examples show generating "opposite" inequalities, but the general principle is not well-articulated.

3. Dataset size limitations: With only 19k problems, the dataset is relatively small compared to other approaches that use hundreds of thousands of examples.

4. Limited theoretical analysis: The paper lacks deeper theoretical understanding of why this approach works better than alternatives. Why do formal statements lead to better reasoning data?

**Questions:**

1. Can you provide more details on how the "underlying logic" of statements is used to generate questions? The current description is quite high-level.
2. How does the approach scale with larger base models? Have you conducted any preliminary experiments with models larger than 8B?
3. How does the performance compare when using equal amounts of data from different sources (e.g., 19k problems from your approach vs. 19k from MATH dataset)?
4. What is the distribution of mathematical domains in your final dataset? Are certain areas over/under-represented?
5.  Do you have any theoretical insights into why training on formal statement-derived problems leads to better generalization?

Computational cost: What are the computational costs of the statement collection and quality control pipeline compared to alternative data synthesis approaches?

---

> ### Author Response · Authors · 2025-11-25
> **Response to Reviewer vBxm (2/4)**
>
> > W2: Unclear problem generation process: The conversion from formal statements to natural language problems could be better explained. The examples show generating "opposite" inequalities, but the general principle is not well-articulated.
> Q1: Can you provide more details on how the "underlying logic" of statements is used to generate questions? The current description is quite high-level.
>
>
> Thank you for pointing this out and we apologize for the confusion. In our problem generation pipeline, (1) the model is instructed to **treat the given statement as implicit background knowledge and then generate questions related to that knowledge**. Solving these questions requires first reasoning to infer the implicit knowledge and then applying it to obtain the answer. (2) when generating ground-truth answers, we deliberately **provide the knowledge explicitly** to easily produce correct answers.
>
> The detailed data generation instructions are provided in **Appendix D.3**. For your convenience, we display the the exact details below:
>
> ```
> [Instruction: Problem Generation Prompt]
>
>
> I have a theorem that has already been proven. Please design an **interesting mathematical problem** based on the given theorem, where the theorem serves as the underlying logic for solving it.
>
>
> The requirements are:
> 1. The generated problem statement **should not reveal the theorem itself** (hide the theorem; do not mention "proof ... then solve" or similar terms). Successfully solving this problem must rely on finding and applying this theorem.
> 2. **When generating problems involving inequality expressions**: Interpret phrases as **bounds/constraints, not exact extrema**. For example, from x ≤ 5 we only know that 5 is an upper bound of x, not that x actually attains the maximum value 5. The fact that such a bound is accepted by the Lean compiler does not imply it represents the maximum or minimum value.
> 3. Only generate problems with a **single, fully determined final answer**. The answer must be uniquely specified (e.g., exact value, complete interval, finite set, Yes/No). Avoid problems where the solution could be partial or admit multiple valid answers.
> 4. You can **analyze and generate its solution directly using the proven theorem to check the answer correctness**.
> 5. The final answer is the summary of the reasoning process; **it can be a numerical value, a range, a short phrase, or a Boolean.** It should be presented in the format: Final Answer: {}.
> ```
>
> Our generation method is general and produces non-trivial questions across diverse mathematical domains beyond inequalities. **To facilitate understanding, we provide the following examples to illustrate how it works in other math domains**:
>
>
> | **Statement** | **Question** | **Answer** |
> |---------------|--------------|------------|
> | Suppose $x$, $y$, and $z$ are real numbers satisfying the following equations: $x - 5 = -(z - 5)(3z^2 + 15z + 19)$, $y - 5 = -(x - 5)(2x^2 + 10x + 13)$, $z - 5 = -(y - 5)(y^2 + 5y + 7)$, and also that $x$, $y$, and $z$ are all non-negative. Then it must be the case that $x = 5$, $y = 5$, and $z = 5$. | Three friends, Alice, Bob, and Carol, are comparing three numbers $a$, $b$, and $c$, each representing the amount of water (in liters) in their individual water tanks. The relationship between their amounts satisfies: $a - 5 = -(c - 5)(3c^2 + 15c + 19)$, $b - 5 = -(a - 5)(2a^2 + 10a + 13)$, $c - 5 = -(b - 5)(b^2 + 5b + 7)$. It is also known that none of the tanks contain a negative amount of water. Determine the amount of water in each tank. | a = 5 liters, b = 5 liters, c = 5 liters |
> | Suppose $f$ is a strictly decreasing function from the real numbers to the real numbers, and it satisfies the property that for all real numbers $x$ and $y$, $f(f(x) + y) = f(y - x) - f(0)$. Then, for all real numbers $x$, $f(x) = -x + f(0)$. | Let $g:\mathbb{R}\to \mathbb{R}$ be a strictly decreasing function such that for some constant $c$, the following relation is satisfied for all real numbers $u$ and $v$: $g(g(u)+v) = g(v-u) - c$. Suppose further that $g(2)=3$. What is the value of $g(10)$? | -5 |
> ||||
>
> We provide additional examples in **Appendix B**, please check. We will also include these cases in Section 2.3 to help readers better understand our generation approach.

---

> ### Author Response · Authors · 2025-11-25
> **Response to Reviewer vBxm (3/4)**
>
> > W3: Dataset size limitations: With only 19k problems, the dataset is relatively small compared to other approaches that use hundreds of thousands of examples.
>
> **Our dataset size reflects a deliberate focus on data quality rather than quantity.** We initially produced approximately **740k** candidate examples, but retained only 19k of them after strict filtering to ensure quality. This aligns with the “less is more” observation in RLVR training, where a smaller set of high-quality samples can outperform large but noisy datasets.
>
> In addition to providing a F1 dataset, **we also contribute a scalable data generation pipeline.** Thus, increasing the dataset size is not a limitation. To demonstrate scalability, we further applied our pipeline to additional statements proven by Goedel-Prover in Leanworkbook, generating 6.5k new QA pairs and forming **F1-Plus**. The performance on Qwen2.5-7B is shown below:
>
> | **Model Name** | **AVG.** | **AMC** | **Minerva** | **MATH** | **Olympiad** | **AIME25** | **AIME24** |
> |---|---|---|---|---|---|---|---|
> | F1-Reasoner | 38.15 | 48.00 |  **52.90** | **74.40** | 38.10 | 3.80 | 11.70 |
> | F1-Reasoner-Plus | **39.32** | **53.00** | 52.60 | **74.40** | **38.70** | **5.40** | **11.80** |
>
>
> We observe that F1-Reasoner-Plus outperforms F1-Reasoner by 1.17 points on Avg, **yielding consistent improvements across the majority of benchmarks, which verifies the scalability of our data generation framework.**
>
> > W4: Limited theoretical analysis: The paper lacks deeper theoretical understanding of why this approach works better than alternatives. Why do formal statements lead to better reasoning data?
> question5: Do you have any theoretical insights into why training on formal statement-derived problems leads to better generalization?
>
> Our primary motivation is to generate correct QA pairs for RLVR. Using seed data from a formal proving system ensures the correctness of the training data. Without such guarantees, the policy model could suffer from two main issues: (1) **Over-optimization** [1]: Optimizing against noisy or incorrect signals can lead the model to reinforce flawed logic merely to maximize the reward resulting in  "reasoning hallucinations". (1) **Shortcut learning** [2][3]: In the presence of noisy data, models tend to memorize reasoning patterns or answer distributions rather than true mathematical reasoning. By generating data in the formal space, we provide a reliable training signal to mitigate these issues, which we believe is the key factor behind the improved reasoning ability.
>
>
> [1] Scaling Laws for Reward Model Overoptimization (ICML, 2023)
>
> [2] Rectifying Shortcut Behaviors in Preference‑based Reward Learning (Arxiv, 2025)
>
> [3] Spurious Rewards: Rethinking Training Signals in RLVR (Arxiv, 2025)
>
> > Q3: How does the performance compare when using equal amounts of data from different sources (e.g., 19k problems from your approach vs. 19k from MATH dataset)?
> Response:
>
> We are sorry for the confusion. Our experiments involve two types of comparisons, each serving a different motivation:
>
> - **F1 vs. other synthetic-environment datasets (e.g., Absolute-Zero, SynLogic)**
>
>  These comparisons aim to evaluate which environment produces higher-quality synthetic data. In these settings, our dataset is significantly smaller (e.g., **SynLogic has 49k examples**), yet our data achieves better performance with fewer samples, suggesting higher data efficiency.
> - **F1-Mix vs. MATH**
>
>  Our Mix dataset is constructed by adding our F1 data in MATH, with the goal of **evaluating whether our synthetic data provides incremental benefits when combined with existing human-curated data.** The results show that Mix consistently improves performance, and the gains are more pronounced for larger models. On Qwen3-8B, F1-Mix even surpasses General-Reasoner-14B, despite the latter being trained on ~230k samples.

---

> ### Author Response · Authors · 2025-11-25
> **Response to Reviewer vBxm (4/4)**
>
> > Q4: What is the distribution of mathematical domains in your final dataset? Are certain areas over/under-represented?
>
> Yes, our approach can generalize to multiple mathematical domains. To analyze the domain coverage of our dataset, we adopt an LLM-as-Judge tagging method to assign each problem to a domain category automatically. After tagging all samples, we examine the distribution across domains. **We find that our dataset spans a wide range of mathematical domains, including areas such as algebra, analysis, discrete mathematics, geometry, and number theory, demonstrating both the domain diversity of the dataset and the generalizability of our generation pipeline.**  We observe that most problems are algebra-related categories. This is likely due to the distribution of formal statements used during data generation. However, this bias is not the  limitation of our method: by sampling formal statements from different domains, we can intentionally control and expand coverage to other areas.
>
>
> | Main Domain | Included Subcategories |
> |-------------|-----------------------|
> | **Algebra** | Algebra, Abstract Algebra, Linear Algebra, Complex Numbers |
> | **Analysis** | Analysis, Mathematical Analysis, Calculus, Complex Analysis, Trigonometry |
> | **Discrete Mathematics** | Discrete Mathematics, Combinatorics, Order Theory |
> | **Geometry & Topology** | Geometry, Topology |
> | **Number Theory** | Number Theory |
> | **Logic & Foundations** | Mathematical Logic, Logic, Set Theory, Mathematical Foundations, Foundations of Mathematics |
> | **Applied Mathematics** | Applied Mathematics, Computer Science, Mathematical Sciences |
> | **Other** | Mathematics, Pure Mathematics, Other |
>
>
> > In summary, we thank the reviewer again for highlighting key points of improvement and opportunities for further work. We hope that these clarifications will lead to a more favorable assessment of our submission.

---

### Official Review · Reviewer_Qhez · 2025-10-31

**Soundness:** 1
**Presentation:** 4
**Contribution:** 1
**Rating:** 2
**Confidence:** 4

**Summary:**

This paper proposes data augmentation by translating formal mathematical statements, e.g., written in lean. The formal statement, verified by the lean compiler, is transformed to a related non-proof problem with a verifiable answer. The dataset, augmented with natural language training data, is utilized for reinforcement learning. Experiments are conducted on Qwen2.5-7B, Qwen3-4B and Qwen3-8B. The proposed method improves by 0.19% over vanilla RL on natural language training data.

**Strengths:**

The paper seeks to address the import research question of synthetic data generation, which is increasingly crucial as models scale and data capacity becomes a bottleneck. The writing is concise while clearly conveys its message.

**Weaknesses:**

1. It is particularly hard to understand the motivation behind translating formal mathematical statements back to natural language. Formalized statements are much more scarce than natural language statements, a significant part of which are actually translated from natural language, such as Lean-Workbook. Augmenting data in the natural language domain is far more flexible than in the formal domain. In fact, formal math provers [1] augment formal statements by synthesizing natural language statements and translating them into the formal domain.

2. The question synthesis stage is not verified to ensure consistency between formal and informal problems. The synthesized informal problem can be far easier than the original formal one. For Example 1 in section D.3, the translated question can be solved by simply plugging in a=0 into the inequality, then confirms that a sufficient small positive 'a' satisfies the inequality for any S < 250. While the original statement requires a proof for arbitrary a>0. For example 3, proof of an inequality is synthesized into deciding whether there is a condition such that the reversed inequality satisfies. This is an yes/no question which can be well guessed even if the solver fails to provide a correct proof.

3. Given that many lean statements are translated from natural language, it is in particular important to examine the improvement of the augmented dataset with respect to natural language datasets. Therefore, the experiment with simpleRL should have been most convincing. However, the result is only provided on Qwen2.5-7B but not for Qwen3-4B and Qwen3-8B. And the improvement on Qwen2.5-7B at 0.19% is not significant enough.

[1] Lin, Y., Tang, S., Lyu, B., Yang, Z., Chung, J. H., Zhao, H., ... & Jin, C. (2025). Goedel-prover-v2: Scaling formal theorem proving with scaffolded data synthesis and self-correction. arXiv preprint arXiv:2508.03613.

**Questions:**

Please refer to weaknesses.

---

> ### Author Response · Authors · 2025-11-25
> **Response to Reviewer Qhez (1/3)**
>
> Thanks for your time and valuable comments. We sincerely appreciate your feedback on F1-Reasoner. We now present a detailed response to address your concerns and comments:
>
> > W1: It is particularly hard to understand the motivation behind translating formal mathematical statements back to natural language. Formalized statements are much more scarce than natural language statements, a significant part of which are actually translated from natural language, such as Lean-Workbook. Augmenting data in the natural language domain is far more flexible than in the formal domain. In fact, formal math provers  augment formal statements by synthesizing natural language statements and translating them into the formal domain.
>
> We sincerely apologize for the confusion. We would like to clarify the motivation and related background behind our work:
>
> - **Motivation**: Our primary goal is to improve the informal mathematical problem-solving ability of large language models. With the success of RLVR, **obtaining reliable QA pairs has become particularly important for reinforcement learning.** Many recent works synthesize data inside artificial environments to construct such correct reasoning QA pairs [1][2]. However, these environments are often far from the real mathematical domain.
> Therefore, we explore **whether it is possible to synthesize QA pairs directly from the formal mathematical environment**, so that the resulting reasoning data are not only genuinely mathematical but also reliable in correctness, which is crucial for RLVR training.
>
>
> - **Source of our statements**: Most of our formal statements come from LLM Conjecturer[3], which **generates new formal conjectures from scratch** rather than translating existing natural-language math questions. These conjectures are then validated by a formal proving system, and only those that pass verification are retained. The resulting verified statements serve as reliable seed data for producing correct QA pairs. **In the final dataset of 19,112 samples, 17,630 are generated from newly synthesized statements, while only 1,482 originate from translated statements.** We believe that ongoing advances of conjecture-generation techniques in the formal math community will yield more diverse and high-quality conjectures.
>
> - **Why do we not synthesize directly from natural language**: Many existing methods [4][5] synthesize math problems directly in the natural-language space. Their typical approach is to let an LLM generate a math problem and then attempt to answer it. This strategy relies on the model’s own reasoning ability to ensure correctness. **When the synthesized problems exceed the model’s capacity, the answers become unreliable.** Such incorrect data can significantly degrade RL performance, as we also observe in Section 3.3 _Training without Formal System Verification._
>
> [1] Absolute Zero: Reinforced Self-play Reasoning with Zero Data (NeurIPS, 2025)
>
> [2] SynLogic: Synthesizing Verifiable Reasoning Data at Scale for Learning Logical Reasoning and Beyond (NeurIPS, 2025)
>
> [3] STP: Self-play LLM Theorem Provers with Iterative Conjecturing and Proving (ICML, 2025)
>
> [4] MathScale: Scaling Instruction Tuning for Mathematical Reasoning (ICML 2024)
>
> [5] R-Zero: Self-Evolving Reasoning LLM from Zero Data (Arxiv 2025)

---

> ### Author Response · Authors · 2025-11-25
> **Response to Reviewer Qhez (2/3)**
>
> > W2: The question synthesis stage is not verified to ensure consistency between formal and informal problems. The synthesized informal problem can be far easier than the original formal one.
>
> Thank you for raising this important concern. Actually, converting a formal theorem into a checkable and informative QA pair is a challenge in the community. **The key challenge is to design a general method that works across diverse math domains while producing meaningful questions that cannot be solved through trivial guessing.**  Prior work primarily follows two directions:
>
> - **Binary proof verification**:
> DeepTheorem[6] converts each theorem into a binary classification task (e.g., determining whether a given  statement is correct). For example:
>
> | **Question** | **Answer** |
> |--------------|------------|
> | Let $f : [a,b] \to \mathbb{R}$ be a continuous function and assume that $f$ is differentiable on $(a,b)$. Prove or disprove the following statement: If $f'(x) = 0$ for all $x \in (a,b)$, then $f$ is constant on $[a,b]$.| Proven |
> |||
>
> This formulation is **highly guessable** and does not require much mathematical reasoning ability.
>
>
> -  **Inequality-focused question generation**:
> IneqMath[7] produces harder QA pairs but restricts itself to inequality problems, such as predicting symbolic forms or extrema. For example:
>
> | **Question** | **Answer** |
> |--------------|------------|
> | Let $a, b, c$ be positive real numbers such that $abc = 1$. Consider the following expressions: $$\frac{b+c}{\sqrt{a}} + \frac{c+a}{\sqrt{b}} + \frac{a+b}{\sqrt{c}} \quad (\quad) \quad \sqrt{a} + \sqrt{b} + \sqrt{c} + 3$$ Determine the correct inequality relation to fill in the blank. Options: (A) $\le$ (B) $\ge$ (C) $=$ (D) $<$ (E) $>$ (F) None of the above | (B) $\ge$ |
> |||
>
> The approach is domain-specific and does not generalize beyond inequalities.
>
> - **Our pipeline is general and produces non-trivial questions across diverse math domains.** These questions are not reducible to binary guessing. For example:
>
> | **Statement** | **Question** | **Answer** |
> |---------------|--------------|------------|
> | Suppose $x$, $y$, and $z$ are real numbers satisfying the following equations: $x - 5 = -(z - 5)(3z^2 + 15z + 19)$, $y - 5 = -(x - 5)(2x^2 + 10x + 13)$, $z - 5 = -(y - 5)(y^2 + 5y + 7)$, and also that $x$, $y$, and $z$ are all non-negative. Then it must be the case that $x = 5$, $y = 5$, and $z = 5$. | Three friends, Alice, Bob, and Carol, are comparing three numbers $a$, $b$, and $c$, each representing the amount of water (in liters) in their individual water tanks. The relationship between their amounts satisfies: $a - 5 = -(c - 5)(3c^2 + 15c + 19)$, $b - 5 = -(a - 5)(2a^2 + 10a + 13)$, $c - 5 = -(b - 5)(b^2 + 5b + 7)$. It is also known that none of the tanks contain a negative amount of water. Determine the amount of water in each tank. | a = 5 liters, b = 5 liters, c = 5 liters |
> | Suppose $f$ is a strictly decreasing function from the real numbers to the real numbers, and it satisfies the property that for all real numbers $x$ and $y$, $f(f(x) + y) = f(y - x) - f(0)$. Then, for all real numbers $x$, $f(x) = -x + f(0)$. | Let $g:\mathbb{R}\to \mathbb{R}$ be a strictly decreasing function such that for some constant $c$, the following relation is satisfied for all real numbers $u$ and $v$: $g(g(u)+v) = g(v-u) - c$. Suppose further that $g(2)=3$. What is the value of $g(10)$? | -5 |
> |||
>
> We have provided additional examples in **Appendix B**, please check.
>
> For the concern about the prompt in section D.3,  **we provide multiple examples in the prompting stage to encourage question diversity.** These examples guide question generation but do not reduce the task to naive guessing, since in  **Example 1**, The question cannot be solved by simply substituting a=0, because the problem explicitly states that a is a positive real number. The value of a is not a constant, and determining whether such an exists requires multi-step reasoning based on the underlying theorem. In our evaluation, DeepSeek-V3.2 took 1760 tokens to solve it. For **Example 3**, the synthesized question is not merely a yes/no check. The prompt asks for a specific value under the hypothetical assumption that such value exists. The correct solution process involves: (i)analyzing the problem, (ii) attempting to construct such a value, and (iii) **concluding that no such value exists, which requires reasoning about why existence fails.**
>
> In summary, while we acknowledge that aligning difficulty between formal statements and informal QA is a complex challenge, our method goes beyond binary verification and inequality-specific formulations, offering a general pipeline that produces diverse and informative questions.  We hope these clarifications address the concern.
>
> [6] Deeptheorem: Advancing llm reasoning for theorem proving through natural language and reinforcement learning (Arxiv, 2025)
>
> [7] Solving Inequality Proofs with Large Language Models (NeurIPS, 2025)

---

> ### Author Response · Authors · 2025-11-25
> **Response to Reviewer Qhez (3/3)**
>
> > W3: Given that many lean statements are translated from natural language, it is in particular important to examine the improvement of the augmented dataset with respect to natural language datasets. Therefore, the experiment with simpleRL should have been most convincing. However, the result is only provided on Qwen2.5-7B but not for Qwen3-4B and Qwen3-8B. And the improvement on Qwen2.5-7B at 0.19% is not significant enough.
>
> Thank you for your advice. In fact, most of our statements are newly synthesized statements generated by an LLM-based conjecturer rather than being translated from natural language, resulting in **17,630 items in the final F1 dataset originating from synthesized statements.** Therefore, such novel mathematical problems serve as a valuable supplement to the existing data.
>
> We also appreciate your observation regarding our Mix version, which is indeed designed to test whether adding our data can further improve the performance of SimpleRL. Since the SimpleRL-Zoo paper only conducts experiments on Qwen2.5 models, **we initially report these results in our paper. However, we have now supplemented the experiments with Qwen3-8B**, and the results are as follows:
>
>
> | **Model Name** | **AVG.** | **AMC** | **Minerva** | **MATH** | **Olympiad** | **AIME25** | **AIME24** |
> |---|---|---|---|---|---|---|---|
> |  | | | Qwen3-8B | | | | |
> | SimpleRL | 53.78 | 75.90 | 52.60 | 88.20 | 51.90 | 23.30 | 30.80 |
> | **F1-Reasoner-Mix** | 54.42 | 72.30 | 66.90 | 87.60 | 55.60 | 20.40 | 25.10 |
>
>
> We observe that F1-Reasoner-Mix improves the average score to 54.42, maintaining **robust performance across general benchmarks while delivering a substantial breakthrough on Minerva-Math (+14.3 points)**.  Considering that Minerva-Math focuses on scientific and quantitative reasoning (e.g., astronomy, physics), **we attribute this improvement to the diverse theorem-based knowledge in our F1 dataset, which effectively enhances the model's generalization to scientific reasoning tasks.** This phenomenon can also be observed in other F1-Reasoner models shown in Table 1.
>
>
> > In summary, we thank the reviewer again for highlighting key points of improvement and opportunities for further work. We hope that these clarifications will lead to a more favorable assessment of our submission.

---

### Author Response · Authors · 2025-11-26
**General Response**

Dear Reviewers:

Thanks for your review and valuable feedback for F1-Reasoner, which make our paper better. As our main contribution, F1-Reasoner offers a general pipeline that produces diverse and informative QA pairs from formal math environment for RL training.  As the formal proving community continues to grow, we believe this direction will become increasingly important and will supply higher-quality data for informal reasoning.

We have revised our paper according to your comments and uploaded it to OpenReview. In addition, for your convenience, we have highlighted the revisions in $\color{orange}{\textbf{orange}}$ in the revised paper. Thank you again for your work and look forward to further communication with you.

best wishes,

Authors of Paper 9641

---

### Author Response · Authors · 2025-12-03
**General Response**

Dear Area Chair and All Reviewers,

We want to express our sincere gratitude for the constructive feedback provided by the reviewers for Submission 9641. We have carefully addressed each of the reviewer's concerns to enhance the quality and robustness of our paper and revised our manuscript. All modifications in the **revised manuscript are highlighted in orange.**  In this General Response, we first briefly recap our contributions, then provide a summary of the rebuttal process, and finally list the key clarifications.

---

## Our Contributions

1. **Data Generation Pipeline**

    ○ We design a scalale and general framework for synthesizing verifiable QA pairs from formal mathematical statements for RLVR training. This offers a solution for utilizing **formal math to enhance LLM informal reasoning** capabilities.

2. **A High-Quality Dataset**

    ○ We present 19k high-quality, challenging math problems covering multiple domains.

3. **Superior Performance**

    ○ Our method achieves significantly better F1-reasoner scores compared to other synthetic data. Furthermore, it serves as an effective supplementary dataset that further boosts strong baselines.

---

## Summary of the Rebuttal Process

We are also grateful that **all discussions and score updates occurred prior to the recent OpenReview information leak**, ensuring that all feedback and evaluations remain objective. During discussion phase, **only 2/5 reviewers provided feedback and they all increased the score of our paper.**

1. **Reviewer JMzx: Rating: 6 $\to$ 8**

   ○ **Reviewer follow-up (26 Nov)**: Reviewer JMzx raised comprehensive questions regarding conceptual clarity, experiments, and writing. We addressed every point and the reviewer explicitly acknowledged our improvements, raising the score to 8.

2. **Reviewer 9RnE: Rating: 4 (Soundness & Contribution increased)**

   ○ **Reviewer follow-up (26 Nov):** Reviewer 9RnE questioned the choice of provers and problem diversity. We integrated the stronger Goedel-prover and demonstrated scalability. The reviewer responded positively, increasing the Soundness and Contribution scores.

3. **Reviewer 7Hrg: Rating: 6**

   ○ **No follow-up**: The main concerns were about the statement sources and data quality. We clarified that ~17k/19k problems are derived from synthetic conjectures and provided details in our data filtering.

4. **Reviewer Qhez: Rating: 2**

   ○ **No follow-up**: We systematically addressed the reviewer's three main concerns (1) **Motivation**: We clarified the data source, providing statistics to demonstrate that the majority of statements are novel synthetic conjectures rather than simple natural language translations. (2) **Problam Difficulty**: We corrected the reviewer's **factual misunderstanding** regarding a prompt example (overlooking the explicit $a>0$ constraint), which leds to an incorrect assumption about problem triviality. Then we provided multiple evidence demonstrating the difficulty of the problems. (3) **Experiments**: We implemented the requested SimpleRL on Qwen3, which further validated the superiority of our method.  **We regret that there was no further communication from the reviewer.**

5. **Reviewer vBxm: Rating: 4**

   ○ **No follow-up**:  We addressed concerns on model scale by adding 14B model experiments despite resource constraints, and provided theoretical analysis. (**Note**: This review was the only one flagged as potentially **Full AI-generated** by https://iclr.pangram.com/submissions).

---

## Key Clarifications

1. **Motivation & Data Source**

   ○ Some reviewers misunderstood our pipeline as translating natural language problems to formal statements and back. We clarified that our statements are primarily newly synthesized conjectures. This aligns with our vision that ongoing advances in formal conjecture generation techniques will yield increasingly high-quality data for our pipeline.

2. **Diversity of Generated Problems**

   ○ To address concerns regarding the generalization of our data generation framework, we provided a detailed breakdown of the mathematical domains within our dataset, along with specific case studies across multiple fields. This demonstrates that our framework is generalizable.

---

By implementing these revisions, we believe the enhancements address the reviewers' initial concerns and provide valuable insights to the field. **Thank you all for taking the time to read our paper and rebuttal. We hope this summary better assists in your evaluation of our work, and we welcome any further suggestions.**

Best regards,

Authors of submission 9461

---

### Meta-Review · Area_Chair_f5fD · 2026-01-13

**Summary:**

The authors clarify that the formal mathematical statements used in this work are primarily generated by LLM-based conjecturers and subsequently verified by formal provers, rather than translated from natural-language problems. This clarification is reasonable and resolves earlier concerns about circularity or reliance on existing natural-language datasets. Using LLM-generated formal conjectures as a scalable source of verifiable mathematical seeds is a sensible and potentially promising direction.

Nevertheless, even accepting this clarification, the reviewers raised substantive concerns that remain unresolved. Chief among these are doubts about whether the generated question–answer pairs faithfully reflect the difficulty and structure of the underlying formal statements, whether the observed performance gains can be causally attributed to the proposed pipeline, and whether the empirical evidence is sufficiently strong and well-controlled. The improvements are often incremental, confounded by data mixing or baseline choices, and not supported by ablations that isolate the key design decisions. As a result, while the paper explores an interesting idea, it does not yet provide a convincing scientific case for its effectiveness or necessity, motivating a rejection recommendation.

**Reviewer Concerns:**

### Concerns Acknowledged / Addressed
- **Source of formal statements**: The authors convincingly clarified that most statements are newly synthesized by LLM conjecturers and verified by formal systems, rather than translated from natural-language problems. This resolves concerns about data circularity and supports the claim that the pipeline can, in principle, scale.
- **Pipeline scalability**: Additional experiments (e.g., with stronger provers and extended datasets) suggest that the statement collection stage is not inherently limited to the initial 19k samples.

### Outstanding Concerns
- **Faithfulness of problem generation**: A central unresolved issue is whether the generated informal QA pairs preserve the semantic difficulty and reasoning demands of the original formal statements. Several examples indicate that the transformation can substantially simplify the task (e.g., reversing inequalities, reducing proof obligations to yes/no existence checks), allowing correct answers without engaging in reasoning comparable to the formal source. Formal verification of the statement does not guarantee that the downstream QA meaningfully reflects that structure.

**Reviewer Scores:**

Reviewer Qhez: 2 → 2

Reviewer vBxm: 4 → 4

Reviewer 9RnE: 4 → 4

Reviewer 7Hrg: 6 → 6

Reviewer JMzx: 6 → 8

---

### Decision · Program_Chairs · 2026-01-26

Reject